# Agent-based and continuous models of hopper bands for the Australian plague locust: How resource consumption mediates pulse formation and geometry

Andrew J. Bernoff[1], Michael Culshaw-Maurer[2], Rebecca A. Everett[3], Maryann E. Hohn[4], W. Christopher Strickland[5], Jasper Weinburd[1]*

**1** Department of Mathematics, Harvey Mudd College, Claremont, California, United States of America, **2** Departments of Entomology and Nematology/Evolution and Ecology, University of California, Davis, Davis, California, United States of America, **3** Department of Mathematics and Statistics, Haverford College, Haverford, Pennsylvania, United States of America, **4** Mathematics Department, Pomona College, Claremont, California, United States of America, **5** Department of Mathematics and Department of Ecology & Evolutionary Biology, University of Tennessee, Knoxville, Tennessee, United States of America

☯ These authors contributed equally to this work.
* jweinburd@hmc.edu

**Data Availability Statement:** All code and numerically generated data files are available from

## Abstract

Locusts are significant agricultural pests. Under favorable environmental conditions flightless juveniles may aggregate into coherent, aligned swarms referred to as hopper bands. These bands are often observed as a propagating wave having a dense front with rapidly decreasing density in the wake. A tantalizing and common observation is that these fronts slow and steepen in the presence of green vegetation. This suggests the collective motion of the band is mediated by resource consumption. Our goal is to model and quantify this effect. We focus on the Australian plague locust, for which excellent field and experimental data is available. Exploiting the alignment of locusts in hopper bands, we concentrate solely on the density variation perpendicular to the front. We develop two models in tandem; an agent-based model that tracks the position of individuals and a partial differential equation model that describes locust density. In both these models, locust are either stationary (and feeding) or moving. Resources decrease with feeding. The rate at which locusts transition between moving and stationary (and vice versa) is enhanced (diminished) by resource abundance. This effect proves essential to the formation, shape, and speed of locust hopper bands in our models. From the biological literature we estimate ranges for the ten input parameters of our models. Sobol sensitivity analysis yields insight into how the band's collective characteristics vary with changes in the input parameters. By examining 4.4 million parameter combinations, we identify biologically consistent parameters that reproduce field observations. We thus demonstrate that resource-dependent behavior can explain the density distribution observed in locust hopper bands. This work suggests that feeding behaviors should be an intrinsic part of future modeling efforts.

a GitHub repository at https://github.com/mountaindust/Locusts_2020 which has been given the DOI 10.5281/zenodo.3738721.

**Funding:** All authors were supported by the Mathematics Research Communities Program of the American Mathematical Society in 2018 under National Science Foundation grant DMS-1321794 (http://www.ams.org/programs/research-communities/mrc). All authors were supported by the Institute for Advanced Study Summer Collaborators Program (https://www.math.ias.edu/summercollaborators). JW is supported by an NSF Mathematical Sciences Postdoctoral Research Fellowship grant DMS-1902818 (https://www.nsf.gov/funding/pgm_summ.jsp?pims_id=5301). WCS was supported by a Simons Foundation Grant #585322. AJB was also supported by a Simons Foundation Grant #317319 (https://www.simonsfoundation.org/grant/collaboration-grants-for-mathematicians/). Funding for open access to this research was provided by the University of Tennessee's Open Publishing Support Fund. The funders had no role in study design, data collection and analysis, decision to publish, or preparation of the manuscript.

**Competing interests:** The authors have declared that no competing interests exist.

## Author summary

Locusts aggregate in swarms that threaten agriculture worldwide. Initially these aggregations form as aligned groups, known as hopper bands, whose individuals alternate between marching and paused (associated with feeding) states. The Australian plague locust (for which there are excellent field studies) forms wide crescent-shaped bands with a high density at the front where locusts slow in uneaten vegetation. The density of locusts rapidly decreases behind the front where the majority of food has been consumed. Most models of collective behavior focus on social interactions as the key organizing principle. We demonstrate that the formation of locust bands may be driven by resource consumption. Our first model treats each locust as an individual agent with probabilistic rules governing motion and feeding. Our second model describes locust density with deterministic differential equations. We use biological observations of individual behavior and collective band shape to identify numerical values for the model parameters and conduct a sensitivity analysis of outcomes to parameter changes. Our models are capable of reproducing the characteristics observed in the field. Moreover, they provide insight into how resource availability influences collective locust behavior that may eventually aid in disrupting the formation of locust bands, mitigating agricultural losses.

## Introduction

Locusts are a significant agricultural pest in parts of Africa, Asia, Central and South America, and Australia. They aggregate in large groups with as many as billions of individuals that move collectively, consuming large quantities of vegetation [1, 2]. Collective movement occurs in both nymphal and adult stages of development and is associated with an epigenetic phase change from a solitary to a gregarious social state which is mediated by conspecific density and abiotic factors [1, 3–6]. Flightless nymphs march along the ground in aligned groups, often through agricultural systems where they cause significant crop damage as they feed and advance [4, 7, 8]. Some species, such as the brown locust *Locustana pardalina*, form intertwining streams of relatively homogeneous density [1, 2, 8]. By contrast the Australian plague locusts *Chortoicetes terminifera* form wide, crescent-shaped bands that contain a high density in front and a rapidly decreasing density behind [4, 9, 10]. Clark [4] notes:

> *The structure of bands varies according to the type of pasture through which they are passing. In areas of low cover containing plenty of green feed, bands develop well-marked fronts in which the majority of hoppers may be concentrated. In areas lacking green feed, bands lose their dense fronts and extend to form long streams, frequently exhibiting marked differences in density throughout.*

As bands of *C. terminifera* move through a field of low pasture, they create a sharp transition from undamaged vegetation in front of the band to significant defoliation immediately behind the band, see a schematic in Fig 1 or aerial photographs such as Figure 2 in [10], Figure 1 in [11], Figure 9 in [12], and multiple images in [13]. In natural systems, *C. terminifera* tend to consume one of several species of grasses; in agricultural systems, they tend to eat primarily pasture and sometimes early stage winter cereals [14].

The Australian plague locust *C. terminifera* is the most common locust species on the Australian continent. For ease, we henceforth refer to *C. terminifera* simply as "locust". Outbreaks of locust nymphs emerge as the result of a pattern of rainfall, vegetation growth, and drought

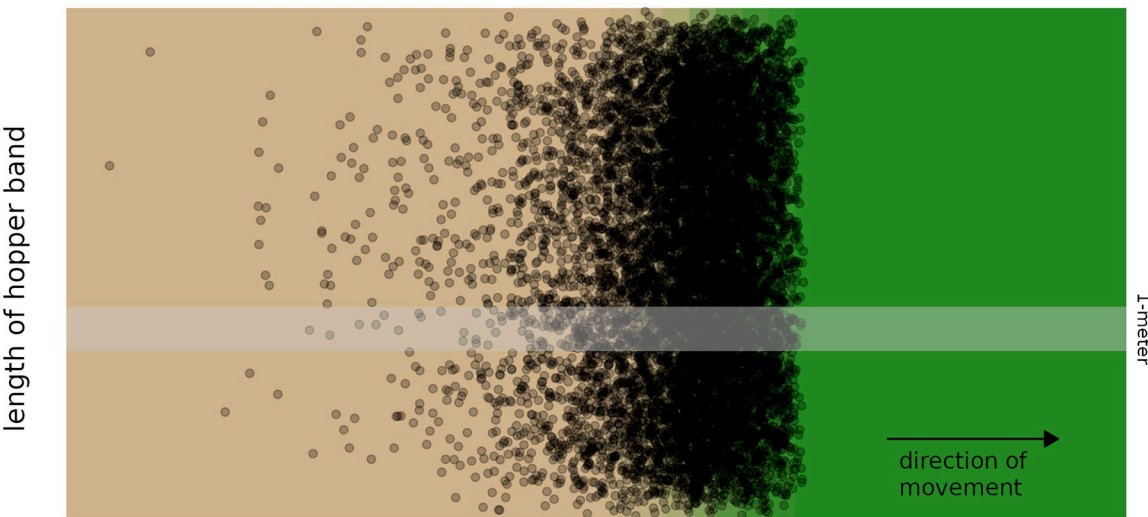

**Fig 1. Schematic of a traveling pulse of locusts.** The Australian plague locust forms broad hopper bands that propagate through vegetation in the direction perpendicular to the aggregate structure [4, 9, 10]. The cross-sectional density profile is a *traveling pulse*, with a steep leading edge (right) and shallower decay behind (left) that is roughly exponentially decreasing in density [9]. Aerial photographs, for instance Figure 2 in [10], show a notable contrast between the verdant green of the unperturbed crops in front of the band and the lifeless brown in the pulse's wake. The one meter wide strip above represents the dimensions we use to model locust movement in a single dimension, as described below.

[11, 15] which promotes breeding, hatching, crowding, and gregarization [3]. Gregarious nymphs form *hopper bands* of aligned individuals, which march distances from tens to hundreds of meters in a single day [4]. Locusts proceed through five nymphal stages, called *instars*, with marching behavior beginning during the second instar [4]. Throughout these phases of life, hoppers consume large quantities of green biomass with an individual eating one third to one half of its body weight per day [14]. Approximately four weeks after eggs hatch, locusts reach adulthood and are then capable of forming even more destructive and highly mobile flying swarms [16].

This study focuses on collective marching in hopper bands, which dominates the behavior of gregarious locust nymphs in the third and fourth instars. Temperature and sunlight dictate a daily cycle of behavior with basking in the morning, roosting at midday, and active periods of collective marching and feeding for up to nine hours when temperatures are in an optimal range ($\sim 25°$C) [10]. During these periods of collective marching, individuals crawl and hop across the ground in nearly the same direction as their neighbors, due to social interactions [17, 18]. When individuals at the front of a band encounter available food resources, they stop and feed (see [4, 10] for qualitative observations and [19] for quantitative experimental results). Immediately after feeding, locusts exhibit a post-prandial quiescent period whose duration increases with the amount consumed [19–21]. Locusts farther back in the band may continue to move forward, eventually passing those that stopped. This creates a "leap-frogging" type motion with a cycling of individuals in the dense front of the band. Clark [4] describes this behavior:

> *Those hoppers behind the front were in places which had been partly or wholly eaten out, and thus lacked the same stimulus of food to stop them. As their average rate of progress was*

*greater than that of the hoppers in the front, they tended to overtake them, becoming in turn slowed down in their progress by the presence of food.*

Thus, individual motion during marching depends on individuals stopping to feed and consequently on local resource density. We hypothesize that this effect mediates the coherence and persistence of hopper bands with a dense front [4, 10] as well as the characteristic cross-sectional density distribution documented in [9].

To test our hypothesis we conduct an in-depth modeling study concentrating on the interaction of pause-and-go motion with food resources. We assume that hoppers march in an aligned band through a field of finite resources, which is depleted as the locusts stop to feed. We develop a model for the probability of movement or stopping as a function of resource availability. We construct and analyze in tandem an agent-based model (ABM), which tracks individual locusts, and a partial differential equation (PDE) model, which considers mean-field densities. Both models produce traveling pulse type solutions that are consistent with the detailed field observations of Buhl et al [9]. The ABM is easily simulated, allows us to track individuals within the swarm, and captures the natural stochasticity of a biological process. In contrast, the PDE produces smooth solutions and lends itself to analysis and a detailed characterization of how observable outcomes, such as mean band speed, cross-sectional density profile, and density of resources left unconsumed in the wake, are related to the model's parameters.

Previous modeling efforts have considered both agent-based and continuous models, see [1] for an excellent overview of locust models. The majority of these have focused on social behavior—notably alignment, attraction, and repulsion with respect to conspecifics [18, 22–27]. Many of the agent-based models consider the pause-and-go behavior of locusts [18, 26, 27], and other insects [28]. Continuous models have been used to study transitions between stationary and moving states [29, 30] and gregarization [31]. Foraging has been modeled in an agent-based framework [32] and resource distribution effects on peak density has been posed as an energy minimization problem [33]. Other continuous models explicitly include food resources having animal movement depend on a combination of aggregation and gradient sensing (chemotaxis in many, starting with [34], or "herbivory-taxis" in [35, 36], for instance). These studies find that traveling animal bands are the result of a balance between attraction to food and inter-animal dispersal, bearing some qualitative resemblance to the results presented here. However, locusts in the present model do not sense resource gradients (instead, direction is prescribed implicitly by social alignment) and the corresponding mathematical equations are distinct from the well-studied equations of chemotaxis.

We are aware of no models of locust band movement that incorporate foraging behavior or food resources. Previous studies such as [23, 27] suggest that the formation of sharp asymmetric fronts may be explained solely by social forces. By contrast, our main conclusion is that foraging and resource-mediated stationary/moving transitions produce pulse-shaped density profiles, supporting the observations of hopper bands with dense fronts and the inferences on foraging of Clark [4] and Hunter et al. [10]. A further strength of our model is that it quantitatively reproduces the observed density profiles of [9] from biologically realistic parameters.

In Models and methods we construct our two models beginning with biological and simplifying assumptions, and ending with parameter identification from empirical field data in Table 2. Our Results describe how both models produce a traveling pulse in locust density precisely when the locusts' stationary/moving transitions are dependent upon the amount of nearby resources. Evidence consists of numerical simulations for the ABM, mathematical traveling wave analysis for the PDE, and a robust sensitivity analysis of the models to changes in the input parameters. In our Discussion we revisit our main findings and outline extensions of

this work incorporating more biological complexity. S1, S2 and S3 Appendices each contain mathematical analysis and proofs substantiating results for the PDE. Finally, S1 Video shows a typical simulation of our agent-based model.

## Models and methods

### Basic assumptions

We outline our assumptions for the modeling framework. Our models are minimal in the sense that we include only the effects necessary to investigate the main question: *Can resource-dependent locust behavior drive the formation of a dense front and the propagation of hopper bands?*

- We assume that resources (food) can only decrease, since locusts feed much more quickly than vegetation grows. Moreover, resources are identical so that they can be characterized by a single variable. Prior to locust arrival, we assume available resources have a spatially uniform density.

- We model only the part of the daily cycle dominated by collective movement. During a typical day, a hopper band has one or two periods of collective movement (marching) totaling up to nine hours. The remainder of the day is spent resting (basking and roosting) [1, 4, 8–10].

- We assume hopper bands consist of flightless nymphs that are behaviorally identical in all regards. Bands often include a mix of two instars (e.g. II and III instars or III and IV instars) which behave qualitatively similarly with later instars being larger, eating more, and moving more quickly.

- We assume individuals move parallel to one another, creating a constant direction of movement for the entire band. Locusts are known to align their direction of movement with their nearest neighbors and may align with environmental cues such as wind or the location of the sun [2, 4, 8].

- We model behavior in a narrow strip aligned to the direction of movement, as shown in Fig 1. For dimensional consistency of the model, we assume the transverse width to be 1 meter.

- We assume that each individual is either stationary or moving. Further, only stationary locusts feed while moving locusts propagate forward with a constant speed that represents an average of crawling and hopping.

- We assume that locusts feed continuously when they are stationary. In fact, locusts eat a meal and then remain sedentary during a post-prandial period [19–21]. While biologically different, these processes are mathematically analogous and we believe including such a delay in the model is unlikely to significantly alter our results.

Furthermore, we make additional assumptions on the rate of transitions between moving and stationary that are supported by empirical observation, although they combine and simplify multiple locust behaviors.

- We assume that locusts transition back and forth between stationary and moving states at a rate depending solely on the resources nearby.

Notably, we have not included any explicit social interaction between locusts; interaction is mediated solely through the consumption of resources. Social interaction plays a well-document role in the aggregation, alignment, and marching of hopper bands, see [1] for instance. By modeling one spatial dimension only, we implicitly include the social tendency of locusts to

align their direction of motion with neighbors as demonstrated in [17]. We do not focus on social interactions simply because our primary goal is to investigate the effect of linking resource consumption with pause-and-go motion on hopper band morphologies.

- We assume the transition rate from moving to stationary is positive and increases as the resource density increases.

Field observations [4, 10] and laboratory experiments [19] have shown that individuals stop marching to eat when they encounter resources in their path. While we assume resources have a uniform local density, the reality on the ground is that a locust is more likely to encounter an edible plant, and thus stop to feed, when the resource density is high.

- We assume the transition rate from stationary to moving is positive and decreasing with resource density.

This behavior is consistent with foraging theories, such as the simple mechanisms illustrated in [37] where insects are likely to leave a patch of resources before the point of diminishing returns. The *Marginal Value Theorem* [38] quantifies this behavior: if an energy cost assigned to foraging is proportional to resource density, then when local resource densities drop below a critical level it costs less energy per unit resource to move on in search of higher density resources. Additionally, there is a second, more subtle behavior behind this assumption. Locusts that become stationary are assumed to have consumed resources. After feeding, locusts exhibit a post-prandial period of inactivity which extends in proportion with the amount consumed [19–21]. Our assumption about this transition rate reflects a longer period of inactivity when resources are plentiful and larger amounts are therefore consumed.

Foreshadowing our results, only one of these two transition rates must depend strictly on local resource availability for our model to produce coherent traveling pulse-type density profiles akin to observed hopper bands with dense fronts.

- These transition processes are completely memoryless, which implies that locusts experience neither hunger nor satiation.

The biological reality is that feeding behavior is complex, see [39] for a review. Locust hunger has been well documented in other species [19, 40]. Since, in our model, locusts are traveling through a field of relatively plentiful resources we suggest that most locusts do not experience starvation (i.e. no sustenance for 24 hours as in some experiments).

We remind the reader that our goal in this study is to demonstrate that resource-dependent behavior is sufficient for the formation and propagation of hopper bands with a coherent dense front. We acknowledge that the efficacy of this model may be improved by adding social interactions—such as alignment, attraction, and repulsion. Additionally, we believe these additions, particularly that of alignment, would play a pivotal role when modeling locust behavior in two-dimensions, as in [23, 27].

## General model formulation

Within the framework described above, we build two models: an agent-based model (ABM) which tracks individual locusts and a partial differential equation model (PDE) that determines locust density. These models share much in their basic structure. Table 1 compares their independent and state variables and Table 2 lists their common model parameters.

**Table 1. Independent and dependent variables appearing in the agent-based and partial differential equation models.** Units are $L$ = length [meters], $T$ = time [seconds], $C$ = number of locusts, $P$ = locust density [number/(meter)$^2$], and $Q$ = resource density [grams/(meter)$^2$].

| Agent-Based Model | Units | Continuous Model | Units | Description |
|---|---|---|---|---|
| $x_n$ | $L$ | $x$ | $L$ | position (along direction of motion) |
| $t_m$ | $T$ | $t$ | $T$ | time |
| $S_{n,m}$ | $C$ | $S(x, t)$ | $P$ | number/density of stationary locusts |
| $M_{n,m}$ | $C$ | $M(x, t)$ | $P$ | number/density of moving locusts |
| $R_{n,m}$ | $Q$ | $R(x, t)$ | $Q$ | edible resource density |

In the ABM, space and time lie on a discrete, evenly spaced lattice $(x_n, t_m)$ while in the PDE space and time $(x, t)$ are continuous. In both models, $S$ and $M$ denote the number or density of stationary and moving locusts respectively. For the ABM, the number of stationary (moving) locusts at $x_n, t_m$ is denoted $S_{n,m}$ ($M_{n,m}$). For the PDE, the analogous continuous quantities for the density of locusts are $S(x, t)$ and $M(x, t)$.

Resources edible by locusts are measured by the non-negative scalar density variable $R$; specifically, the resource density in the agent-based model is $R_{n,m}$, and the resource density in the continuous model is $R(x, t)$.

We assume that the group rate of feeding is proportional to the product of the stationary locust density and the resource density; that is,

$$(\text{rate of change of resources at a given location}) = -\lambda S R \tag{1}$$

where $\lambda$ is a positive rate constant that describes how quickly individual locusts consume resources. This implies that an individual locust's foraging efficiency decreases as resources become scarcer at their location. This is not an explicit implementation of the Marginal Value Theorem but fits the general concept of foraging efficiency within a patch decreasing due to searching time, not satiation by the forager [38]: as the resources at a location are eaten, locusts have difficulty locating the next unit to consume, reducing the overall rate of resource consumption at that location. We will refer to $\lambda$ as the *foraging rate*, as it reflects both feeding and foraging efficiency.

We model the stationary-moving transitions as a Markov (memoryless) process. For the PDE model, this yields a rate at which the population of stationary locusts transitions to moving, and vice versa. This assumption ignores the transition history and hunger (as discussed above) of any individual locust, which is justifiable on the timescale of the collective motion

**Table 2. Estimates of biological parameters for both models.** Parameters above the horizontal line are estimated from empirical observations, with explanations in text. Parameters below the horizontal are estimated from collective information and model behavior. Units are $L$ = length [meters], $T$ = time [seconds], $C$ = number of locusts, $P$ = locust density [number/(meter)$^2$], and $Q$ = resource density [grams/(meter)$^2$].

| | Description | Units | Min | Max | Example | Source |
|---|---|---|---|---|---|---|
| $N$ | total number locusts in strip | $C/L$ | 5000 | 30000 | 7000 | [9] |
| $R^+$ | resource density in front of band | $Q$ | 120 | 250 | 200 | [44] |
| $v$ | individual marching speed | $L/T$ | 0.003 | 0.1 | 0.04 | [27, 45] |
| $\alpha$ | $S \rightarrow M$ transition rate for $R = 0$ | $1/T$ | $\eta$ | 1 | 0.0045 | [27] |
| $\beta$ | $M \rightarrow S$ transition rate for $R = 0$ | $1/T$ | 0.01 | $\theta$ | 0.02 | [26] |
| $\eta$ | $S \rightarrow M$ transition rate, large $R$ | $1/T$ | 0 | $\alpha$ | 0.0036 | |
| $\theta$ | $M \rightarrow S$ transition rate, large $R$ | $1/T$ | $\beta$ | 12.5 | 0.14 | |
| $\gamma$ | exponent of $S \rightarrow M$ transition | $1/Q$ | 0.0004 | 0.08 | 0.03 | |
| $\delta$ | exponent of $M \rightarrow S$ transition | $1/Q$ | 0.0004 | 0.08 | 0.005 | |
| $\lambda$ | individual foraging rate | $1/TP$ | $10^{-10}$ | $10^{-4}$ | $10^{-5}$ | [46] |

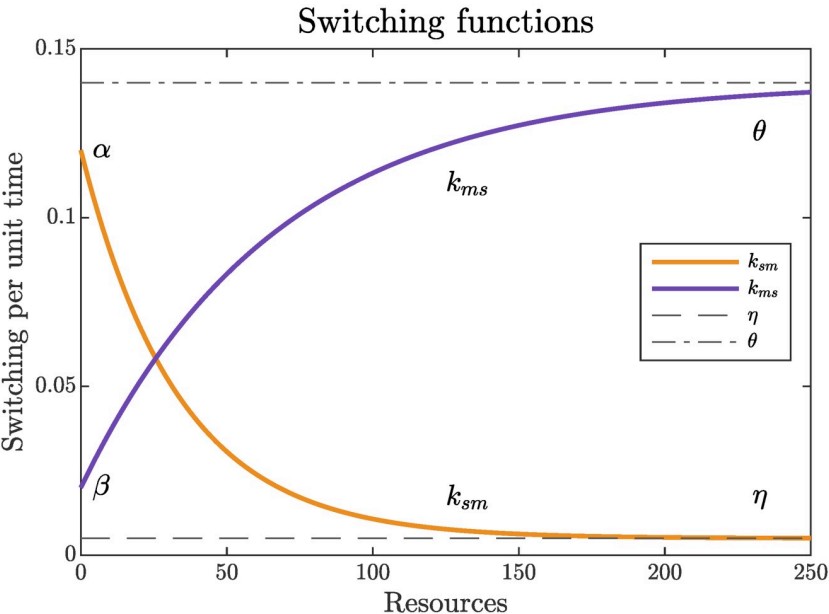

**Fig 2. Transition rates for stationary to moving $k_{sm}$ (gold) and moving to stationary $k_{ms}$ (purple) with $\alpha$ = 0.12, $\beta$ = 0.02, $\gamma$ = 0.03, $\delta$ = 0.015, $\eta$ = 0.005, and $\theta$ = 0.14.**

(hours). We use exponentially saturating functions of resources as illustrated in Fig 2. The stationary to moving rate is denoted $k_{sm}$ while the moving to stationary rate is called $k_{ms}$. Specifically,

$$k_{sm}(R) = \eta - (\eta - \alpha)e^{-\gamma R}, \qquad k_{ms}(R) = \theta - (\theta - \beta)e^{-\delta R}, \qquad (2)$$

where $\gamma, \delta > 0$, $0 \leq \beta \leq \theta$, and $0 < \eta \leq \alpha$. The conditions on the parameters guarantee that $k_{sm}(R)$ is a decreasing function and $k_{ms}(R)$ is an increasing function of $R$. (Most of the analytical results concerning the PDE model hold for any choice of monotone switching rates—see S2 Appendix for details).

This functional form derives from the assumption that the transition rate's sensitivity to changes in resources is proportional to the resource availability [38]. Biologically, this implies that when encountering excess resources, there will be a high proportion of stationary locusts, and doubling the excess resources will do little to change the proportion of stationary locusts. Similarly, when resources are scarce, locusts are most likely to transition from stationary to moving and least likely to stop. Mathematically, this functional form preserves the positivity of the transition rates and means that the transition rates are constant in the limit of abundant resources.

In the PDE model, the transition rates $k_{sm}$, $k_{ms}$ appear as coefficients in growth and decay terms in the differential equations. In the ABM we use a stochastic version of these transitions. At each time step, locusts switch from stationary to moving via a transition probability $p_{sm}$ and from moving to stationary via $p_{ms}$, both of which are functions of $R_{n,m}$. The smooth transition rates $k_{ms}$ and $k_{sm}$ can be understood to be derived from these probabilities as the time step $\Delta t$ approaches zero. Assuming $\Delta t$ is small yields the following approximations,

$$p_{sm}(R_{n,m}) \approx k_{sm}(R_{n,m})\Delta t, \qquad p_{ms}(R_{n,m}) \approx k_{ms}(R_{n,m})\Delta t. \qquad (3)$$

This is equivalent to assuming that each locust undergoes only a single transition in any given time step. Biologically, these transition probabilities can be estimated from intermittent motion observed in the laboratory [26, 41] or the field [18, 27]. These observations suggest that transitions occur on a timescale of a few second. Additionally locusts also exhibit a post-prandial quiescence which may last several minutes, particularly after a large meal [19–21]. These timescales are much shorter than the period of collective marching (hours) which justifies our original approximation that the process is Markovian (memoryless).

## Agent-Based Model (ABM): Pause-and-go motion on a space-time grid

We now describe the details and implementation of our agent-based model (ABM) which encodes the behavior of each individual locust. The temporal evolution of the ABM may be thought of as a probabilistic cellular automaton. The model is one dimensional in space, representing a 1-meter-wide cross section of the locust hopper band.

Our ABM tracks the position of each locust, their states (stationary or moving), and the spatial availability of resources (food). Locust position and the spatial distribution of resources are confined to a discrete lattice of points given by $x_n = n\Delta x$ and time $t_m = m\Delta t$, for $n, m \in \mathbb{N}$. We fix $\Delta x = v\Delta t$ so that a moving locust moves forward one step on the lattice per each time step.

Let $X_i(t_m)$ be the position of the $i^{\text{th}}$ locust at time $t_m$. Let $\sigma_i(t_m)$ be a binary state variable where $\sigma_i = 1$ when the locust is moving and 0 otherwise. The motion of the locusts can now be expressed succinctly as

$$X_i(t_{m+1}) = X_i(t_m) + \sigma_i(t_m)v\Delta t = X_i(t_m) + \sigma_i(t_m)\Delta x, \tag{4}$$

where we have applied the value of the state variable at $t_m$ throughout the interval of length $\Delta t$. Note this artifice ensures that the values $X_i$ remain on the lattice for all time $t_m$.

We model transitions between stationary and moving states with a discrete-time Markov process given via the probabilities in Eq (3). Thus, at time $t_m$, each locust at position $x_n$ has a probability $p_{sm}(R_{n,m})$ to switch from stationary $\sigma_i = 0$ to moving $\sigma_i = 1$ or a probability $p_{ms}(R_{n,m})$ to switch from moving to stationary.

We define the histogram variables mentioned above by simply counting the number of locusts in each state at each space-time grid point:

$$M_{n,m} = \sum X_n(t_m)\sigma_n(t_m) = \#\ \text{of moving}\ (\sigma_i = 1)\ \text{locusts at}\ (x_n, t_m) \tag{5}$$

$$S_{n,m} = \sum X_n(t_m)(1 - \sigma_n(t_m)) = \#\ \text{of stationary}\ (\sigma_i = 0)\ \text{locusts at}\ (x_n, t_m). \tag{6}$$

We model the resources with a scalar variable $R_{n,m}$ which is defined as available food, measured in grams, at time $t_m$ in the interval of width $\Delta x$ centered at $x_n$. Following Eq (1) and converting $S_{n,m}$ to a density, we have

$$\frac{dR_{n,m}}{dt} = -\lambda \frac{S_{n,m}}{\Delta x} R_{n,m}. \tag{7}$$

Solving Eq (7) (assuming $S_{n,m}$ is constant between $t_m$ and $t_{m+1}$) yields

$$R_{n,m+1} = R_{n,m} e^{-\lambda S_{n,m}\frac{\Delta t}{\Delta x}}. \tag{8}$$

Biologically, this evolution implies that the resources in a patch of vegetation infested by a group of stationary, feeding locusts will decrease by approximately half in an amount of time inversely proportional to $\lambda$ times the number of locusts in the patch. That is, the half-life of

resources in the patch is $\ln(2)\Delta x/(\lambda \cdot \# \text{ locusts})$. We initialize each simulation with $R_{n,1} = R^+$, indicating an initially constant field of resources.

Together with initial conditions, Eqs (3), (4) and (8) specify the evolution completely. Our agent-based model then takes the form of three sequential, repeating steps for each locust agent:

1. Update state $S$ or $M$ according to the Markov process.

2. If in state $M$, move to the right $\Delta x$.

3. If in state $S$, decrease resources in current location.

Each locust performs each of these steps simultaneously with all other locusts, and resources in each location are also updated simultaneously according to Eq (8).

## PDE model: A conservation law for locusts

We construct a continuous-time, mean-field model for the density of locusts. As outlined in General model formulation, we write a continuous function of space and time $R(x, t)$ for the density of available resources. Similarly, we write $S(x, t)$ and $M(x, t)$ for the density of stationary and moving locusts, respectively. See Table 1 for comparison with the variables of the agent-based model.

These densities are governed by the partial differential equations

$$\begin{aligned} R_t &= -\lambda S R \\ S_t &= -k_{sm}S + k_{ms}M \qquad\qquad x \in \mathbb{R}, \quad t \in [0, \infty), \qquad (9) \\ M_t &= k_{sm}S - k_{ms}M - \nu M_x \end{aligned}$$

which describe the feeding, switching, and movement behaviors on the scale of the aggregate band. The rate of decrease of $R$ is proportional to the density of stationary locusts and available resources as established in General model formulation. The constant of proportionality is given by the foraging rate $\lambda$. As in the ABM, locust foraging efficiency decreases as resources decrease. Note that the food $R$ is decreasing in time at each spatial point $x$. The rate of change of $S$ is determined wholly by the switching behavior. Here, the decrease of $S$ represents the switching of locusts from stationary to moving with a rate dependent on $R$ through $k_{sm}(R)$. Similarly, $S$ increases as locusts switch from moving to stationary with rate $k_{ms}(R)$. See Eq (2) for the functional forms of $k_{sm}$, $k_{ms}$. The same terms with opposite signs contribute to changes in $M$. The term $\nu M_x$ in the equation for $M$ represents the marching of moving locusts to the right with the individual speed $\nu$. This spatial derivative makes the third equation into a standard transport equation. A full list of all parameters appears in Table 2.

We consider initial conditions with resources that are a positive constant $R^+$ for large $x$; that is, $R(x, 0)$ has $\lim_{x \to \infty} R(x, 0) = R^+$. We assume initial locust densities $S(x, 0)$, $M(x, 0)$ are non-negative and smooth (continuous with continuous derivative). For biologically reasonable choices of such initial conditions, all solutions are guaranteed to remain non-negative, continuous, and finite by standard quasilinear hyperbolic PDE [42].

Finally, since the switching terms are of opposite signs in the $S$ and $M$ equations, we have mathematically guaranteed a conservation law. In particular, the total number of locusts in our 1-meter cross section $N = \int_{-\infty}^{\infty} (S + M)\, dx$ is conserved.

**Numerical simulations.** For direct numerical simulations of the PDE, we use a 4th-order Runge-Kutta method for the temporal derivative with step $dt$. By choosing $dx = \nu \cdot dt$ we approximate the spatial derivative by a simple shift of the discretized $M$ on the spatial grid.

**Table 3. Collective observables with ranges based on field research.** Units are $L$ = length [meters], $T$ = time [seconds], $C$ = number of locusts, $P$ = locust density [number/(meter)$^2$], and $Q$ = resource density [grams/(meter)$^2$]. Note that skewness ($\Sigma$) is nondimensional.

| Symbol | Description | Units | Min | Max | Example Output | Citation |
|--------|-------------|-------|-----|-----|----------------|----------|
| $c$ | speed of collective band | $L/T$ | 0.0005 | 0.009 | 0.0053 | [4, 10] |
| $R^-$ | remaining resource density | $Q$ | 0 | 100 | 0.002 | [44] |
| P | maximum locust density | $P$ | 950 | 4280 | 1296 | [9, 10, 47] |
| $W$ | threshold width of profile | $L$ | 30 | 500 | 18.6 | [9, 10, 47] |
| $\Sigma$ | skewness of locust profile | 1 | 1 | 2 | 1.78 | [9] |

This is equivalent to a first-order upwind scheme because

$$\frac{M(x_n, t_{m+1}) - M(x_n, t_m)}{dt} = -v\frac{M(x_n, t_m) - M(x_{n-1}, t_m)}{dx} \Rightarrow M(x_n, t_{m+1}) = M(x_{n-1}, t_m).$$

For additional accuracy, we implement these schemes using a split-step method, as in [43] for instance. All simulations of the PDE used Matlab.

## Parameter identification

We identify a range of values for biological parameters from a variety of sources including research papers, Australian government guides and reports (particularly the Australian Plague Locust Commission), and agricultural organizations. A list of input parameters and ranges can be found in Table 2. A list of observable outcomes can be found in Table 3.

**Input parameters.**

*Empirical estimates.* We estimate five parameters directly from empirical observations: the total number of locusts $N$ in the cross section, the initial resource density $R^+$, the speed $v$ of an individual locust, and the two switching rates when no resources are present $k_{sm}(R = 0)$, $k_{ms}(0)$. We provide ranges for these parameters in the first five rows of Table 2.

The total number of locusts $N$ in our model is the number of locusts in a 1-meter cross section as shown in Fig 1. We rely on Buhl et al [9] to estimate $N$. In Figure 1 of [9], the authors present three profiles of locust density computed by counting locusts in frames of video of a marching locust band taken during field experiments. The authors fit exponential curves to these data, see Figure 2 in [9], which yield exponential rates of decay of density in time. We use these rates to estimate the area under the density profiles by integrating a corresponding exponential function. This provides three estimates for the total number of locusts who passed under the camera, which range from 9300 to 15000. Rather than a precise measurement, we consider this an estimate and acknowledge that it may be improved by more direct analysis of the underlying data in [9]. We believe it does capture the correct order of magnitude and so include only a modestly larger range in our table.

Typical resource densities $R^+$ come from Meat and Livestock Australia [44]. This resource indicates that pasture with vegetation between $4 - 10$cm high is desirable for livestock grazing. It also converts this range to a vegetation density measured in units of kilograms green Dry Matter per hectare. (Note that this measure discounts the mass of water in the vegetation, sometime up to 80%. While locusts typically feed on live non-dry vegetation, its water content does not provide energy or nutrients. As a result our variable $R$ reflects not the harvestable greenery but instead represents the locust-edible resources.) We convert units and arrive at the range given in our table.

We obtained the speed $v$ of an individual marching locust from experimental measurements in [45] and field data reported in [27]. The experiments were conducted with the desert

locust, *Schistocerca gregaria*, and results in a range of 0.0339 − 0.0532 m/sec. This range contains the estimate from field data for the Australian plague locust collected by Buhl and and reported by Bach [27]. The latter source also provides a second (higher) estimate that accounts for hopping, a common behavior of the Australian plague locust. Buhl's observations also show an increase in an individual's speed (averaged over crawling and hopping) with increasing temperature. Our range in Table 2 spans all of these estimates. Most other recorded observations of speed represent collective information—the speed of the aggregate band—which we discuss in the subsection Collective observables—model outcomes. below.

Constants $\alpha$ and $\beta$ represent the proportion of locusts that switch from stationary to moving (and vice versa) on bare ground, $R \approx 0$. One laboratory study [26] with *S. gregaria* provides data from which we draw out a single estimate for $\beta$ as follows. The authors record the probability of these transitions in a laboratory area with no food present. They construct probability distributions (depending on time) for these transitions and fit curves to these distributions, see Figure 1 in [26]. They find an exponential best fit for the probability that a locust transitions from moving to stationary. The exponential rate represents a reasonable value for $\beta$, so we gather that $\beta \approx 0.368$ sec$^{-1}$. We use this estimate to set a minimum value of $0.01 < \beta$ and provide an upper limit below. The same source does not provide an estimate for $\alpha$ because the authors find that the probability distribution for stationary to moving transitions is best described by a power law.

Instead, for $\alpha$ we rely on the field data of Buhl appearing in [27] for *C. terminifera*. A similar procedure as above yields an estimate of $\alpha \approx 0.56$ sec$^{-1}$. We use this to set a maximal value of $1 > \alpha$ and provide a lower limit below. In using ranges for $\alpha$ and $\beta$, we aim to allow for natural variation between the two species for which there is data.

*Additional parameters*. The parameters below the horizontal line in Table 2 do not all have readily available estimates in the literature; likely because the individual information encoded in these parameters is difficult to measure empirically amid the chaos of the swarm. We discuss the effects each in our Parameter sensitivity analysis.

Constants $\eta$ and $\theta$ represent the proportion of locusts that begin/restart or stop marching in a resource-rich environment, $R \approx R^+$. To empirically measure these would require a detailed examination of locusts marching in natural plant cover. We are not aware of a situation where such a study of marching has been conducted in a setting with abundant food.

To choose a range for $\eta$ we rely on our biological assumption that a locust is more likely to begin moving when there are fewer resources nearby; that is, $\eta < \alpha$. (In our Parameter sensitivity analysis, this results in the bound $\eta/\alpha < 1$.) This assumption provides a lower bound for $\alpha$ and an upper bound for $\eta$. We choose 0 as a lower bound for $\eta$, since it seems conceivable that a hungry locust might be satisfied to remain near food indefinitely. The converse biological assumption, that a locust is less likely to stop moving when there are fewer resources nearby, leads us to conclude that $\beta < \theta$. (In our sensitivity analysis of, this results in the bound $1 < \theta/\beta$.) This provides our upper limit for $\beta$ and our lower bound for $\theta$. We choose our upper limit for $\theta$ to be significantly larger than $\eta$, the comparable transition rate with nearby food. This encodes an assumption that the attraction of nearby food is stronger than its absence. Note that these bounds are contained in the conditions we listed after introducing $k_{sm}$, $k_{ms}$ in Eq (2). Namely, these choices force the transition rates to be decreasing and increasing respectively.

The parameters $\gamma$ and $\delta$ determine how sharply the transition rates $k_{sm}(R)$ and $k_{ms}(R)$ depend on resources $R$. Specifically, they are the rate of exponential decrease and increase, respectively. One of our primary claims is that $\gamma$ and $\delta$ must be positive, otherwise the transition rates $k_{sm}$ and $k_{ms}$ would be constant. More specifically, one may deduce that $\gamma, \delta$ should be of the same magnitude as $1/R^+$, since the functions $k_{sm}(R)$ and $k_{ms}(R)$ are defined on the

interval $[0, R^+]$. Using our range of $R^+$ values above, we obtain the ranges appearing in the table for $\gamma$ and $\delta$.

The individual foraging rate $\lambda$ is difficult to estimate for two reasons. First, it represents an instantaneous rate of change while most data on locust consumption is averaged over days or weeks, as in [47]. We found finer measurements of feeding in [46], where rates are averaged over ten-minute intervals. After unit conversions, we estimated a range of consumption rates on the order of $10^{-8} - 10^{-6}$ grams/(locust·sec). However, these rates are measured in a laboratory setting where locusts are provided with abundant resources to feed. This highlights a second difficulty in estimating $\lambda$; the lab data does not account for search times and so may represents a "consumption rate" rather than a foraging rate. To explain, recall that our ABM places a locust at a grid point which represents a rectangle of physical space with dimensions $\Delta x \times 1$ m$^2$. A locust may need to move within this small rectangle to find an individual plant suitable for feeding. Since we track only the resource density in that local rectangle, this search time is simply accounted for in the foraging rate. Other factors such as digestion times and the post-prandial rest period complicate the matter further. With such persistent uncertainty, we allow a large range for $\lambda$ and explore it thoroughly in our Parameter sensitivity analysis.

*Example values*. Throughout the remainder of the text we illustrate our results using the set of example parameter values appearing in the second column from the right in Table 2. These values produce in both models a density profile consistent with observed locust bands. We selected these values using insight gleaned from our parameter sensitivity analysis, for details see the end of our Parameter sensitivity analysis.

**Collective observables—Model outcomes.**   We consider five measures of collective behavior. Table 3 provides an empirical range for each, estimated in the following paragraphs from data in the literature.

We approximated the collective speed $c$ of the band from observations in [4, 10]. Authors of [10] observed that bands moved between $36 - 92$ meters per day (in "green grass"). Table 4 in [10] estimates the times of day during which marching was observed, with a range of 3 to 7 total hours per day. We computed averages over these time intervals and converted units to obtain a range. In Clark [4], bands of locusts were observed for periods of an hour during daily marching and reports a range of average band speeds overlapping with the range computed above. Our Table 2 shows the union of all three ranges with rounding. Measuring this observable in our models is straightforward. In simulations of either the ABM or PDE we compute the mean position (or center of mass) of the locust band. Tracking the speed of the center of mass gives us the mean speed of the band. Additionally, analysis of the PDE model yields an explicit formula of for $c$ with no need for simulations, details in Theoretical results for the PDE: Hopper bands as traveling waves.

The density of locust-edible food resources left behind by a band $R^-$ does not appear to be well studied. Wright [47] makes a careful study of leftover grain fit for human consumption; however, data are reported after threshing and processing and does not describe the amount of remaining green matter edible by locusts. An alternative approach to understanding $R^-$ could be to use [44] which suggests that a low range of green dry matter in pastures is $40 - 100$ grams per square meter. This low range of green dry matter inhibits vegetation regrowth, increases erosion hazards, and is insufficient for grazing livestock. We emphasize that there is no data suggesting that a marching locust band leaves a field with leftover vegetation in this range. In particular, this provides us with an upper range only since some of the vegetation left behind may be inedible, even for voracious locusts. Thus we arrive at a lower bound of zero for $R^-$. To measure the resources left behind in our models, we take a spatial average over the part of our domain to the left of the band of locusts.

The maximum locust density P = max($S+ M$) in a band is taken from Table 1 in [10]. We used the range of estimates observed for III and IV instars. This range is in line with the data of [47] who estimated a maximum density of 4000 locusts per square meter. In [9], the authors observed maximum densities ranging from $600 - 1200$ locusts per square meter. We expect that these densities lie in (and just out of) the lower end of our range because the studies of [9] were conducted on bare ground with no vegetation while, typically, locusts aggregate into denser bands in lush vegetation, as observed in [4] for instance. The maximum density of a band in our models is measured simply by adding the components $S$ and $M$ and taking the maximal value.

The width $W$ of the band, measured parallel to the direction of motion, is taken primarily from Hunter et al [10]. Hunter et al measured the widths of bands by walking from the front into the band until "marching was no longer seen". Estimates from other sources fall in line with part of the range found in Hunter. For instance, $30 - 140$ meters in [47] or $50 - 200$ meters in [9]. We attribute the large range of band widths in [10] to the fact that these observations come from bands with a variety of sizes, as can also be seen by the large range for maximum densities in the same data set.

Measuring band width $W$ in our model is not entirely straightforward as we cannot simply observe where "marching [is] no longer seen", as in [10]. Marching refers to a consistent movement of locusts with a preferred direction determined by alignment with their nearby neighbors. Since our models assume that locusts are always highly aligned, we rely on the locust density to determine where marching occurs. Experimental data and modeling work in [17] suggest that locusts in a group with a density greater than 20 locusts per square meter are likely to be highly aligned. We thus take $W$ to be the length of the spatial interval where our density profile measures above the threshold of 20 locusts/m$^2$.

This threshold definition of width $W$ is biological and observable but it is not a good quantitative measure of the shape of a density distribution. For instance, consider a distribution with a maximum density less than the threshold density. This distribution will always measure $W = 0$ regardless of if it is very wide with a large total mass or if it is narrow with a much smaller mass. In other words, $W$ does not scale with the total number of locusts in our band. We therefore introduce a second notion of width for use in comparing the shapes of bands with different total masses. A natural choice is the standard deviation of locust positions. We denote our standard deviation width by $W_\sigma$ and use it particularly in our Parameter sensitivity analysis. Unfortunately, there is no general correspondence between our two notions of width $W$ and $W_\sigma$. Even for a fixed mass, one can construct distributions with different shapes and broad ranges of $W_\sigma$ while keeping $W$ constant. For a given parameter set and varying mass we do compute $W$ and make some a posteriori comparisons below.

The skewness $\Sigma$ of a distribution is the third central moment (nondimensionalized by $W_\sigma^3$) and measures the distribution's symmetry about its mean. When $\Sigma = 0$ the distribution is symmetric while $\Sigma > 0$ suggests the distribution is leaning to the right with a longer tail on the left. (We acknowledge that this is the opposite of the standard convention.) Any exponential distribution $e^{-Ax}$ has skewness $\Sigma = 2$. Since [9] has demonstrated that an exponential fits well the locust density behind the peak, we consider 2 as a physically realistic upper bound. Including the sharp increase and maximum density at the front of the band will decrease skewness suggesting that we might expect values in the range $1 < \Sigma < 2$.

*Collective observables for example values.* The example parameter values produce rather realistic collective outcomes; each of them is very nearly in the range obtained from the literature, see Table 3. A small exception is the threshold width $W$, which is less than twelve units outside a large range of several hundred units. Secondarily, we remind the reader of our

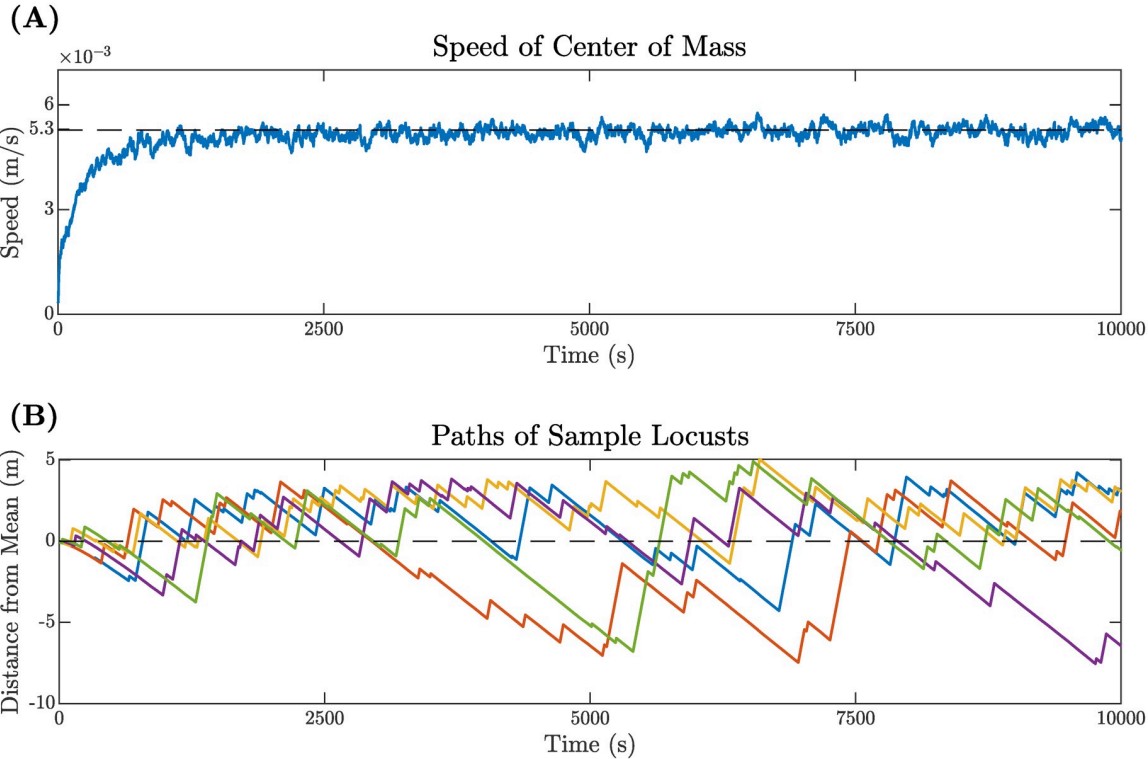

**Fig 3.** (A) Speed of the mean position of all locusts (center of mass of the swarm). Note the initial increase followed by a sustained period of variation around the average $c = 5.3 \times 10^{-3}$ m/sec. The standard deviation around $c$ is $0.16 \times 10^{-3}$ m/sec after transients. (B) Paths of five sample locusts, each shown in a different color. Note the initial transients appearing as curves near $t = 0$, after which all each path appears piece-wise linear with either positive or negative slope corresponding to when the given locust was in a moving or stationary state. Each locust spends some time ahead of the mean and some time behind it, reminiscent of the "leap-frogging" behavior noted in [4].

difficulty in estimating the remaining resources. We interpret the small value $R^- = 0.002$ g/m$^2$ to mean that in our models bands of locusts eat essentially all of the edible vegetable matter. We do not claim that they leave behind no vegetation at all.

## Results

### Numerical results for the ABM

Typical behavior for the agent-based model is a transient period followed by a traveling pulse shape, see S1 Video for a typical simulation. During the transient period, the locust histogram variables $S_{n,m}$ and $M_{n,m}$ evolve to an equilibrium profile that moves with constant speed, each with stochastic variation at each time step. The duration of the transient period and shape of the equilibrium profile vary depending on biological input parameters, while the level of stochastic noise depends primarily on the size of $\Delta t$. We explored a refinement of $\Delta t$ from 1 sec to 0.1 sec and observed similar behavior with decreasing levels of noise. In all results presented in this section we use $\Delta t = 1$ sec and our example values from Table 2 for all biological parameters.

Fig 3A shows the instantaneous speed of the mean position of all locusts over the course of 10000 sec. After an initial increase, the speed stabilizes around an average $c = 5.3 \times 10^{-3}$ m/sec with a standard deviation of $0.16 \times 10^{-3}$ m/sec. Individual locusts move according to a biased

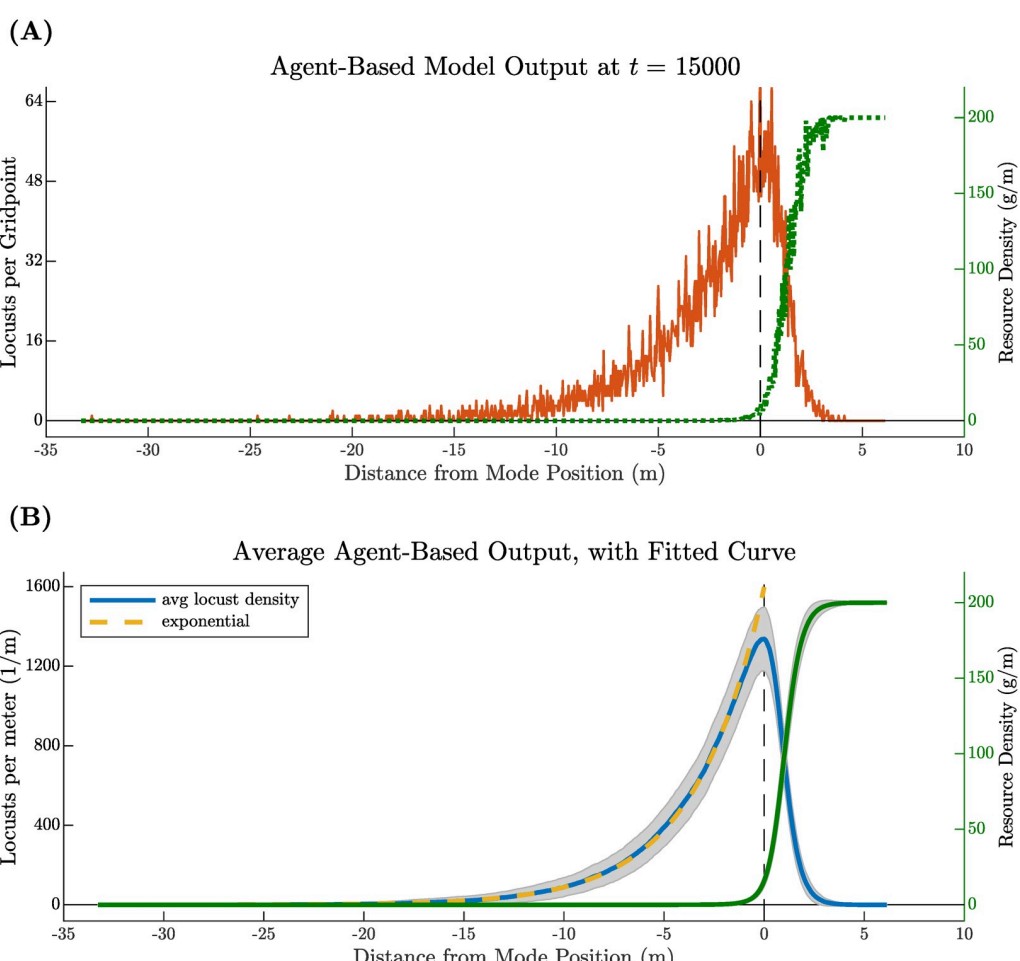

**Fig 4. Output of the agent-based model with *N* = 7000 locust agents at time *t* = 15000 sec, re-centered so that the mean position of all locusts occurs at zero.** (A) Shows the final state of the model including the number of locusts at each spatial grid point (orange) and the remaining resource density at each spatial grid point (dotted, green). Compares well with previously published data, see Figure 1 in [9]. (B) Displays a time-average of model outputs taken after an amount of time to account for transients (in this case, approximately *t* = 7500). Gray shading indicates ± one standard deviation from the average locust density (blue) and resources (green). The tail of the pulse agrees well with an exponential least squares fit (gold). S1 Video shows a full, time-dependent simulation of the ABM.

random walk around the mean position, as illustrated by the paths of five sample locusts in Fig 3B. Note the brief period of transients visible as arcing curves near *t* = 0, after which the distance to the mean is given by a piece-wise linear function for each locust. Intervals with positive slope *v* − *c* correspond to periods where the individual was moving, while negative slope −*c* indicates periods where the individual was stationary and the mean position marched on ahead.

The shape of the traveling pulse may be seen in Fig 4. The final histogram of locusts per spatial grid point at time *t* = 15000 appears in Fig 4A. A time-averaged pulse shape appears in Fig 4B. We construct this smooth density profile by averaging histograms for all time steps after the end of transients, in this case approximately *t* = 7500. Both plots show corresponding resource levels. The resources left behind $R^-$ after the pulse has completely passed depends primarily on the foraging rate constant λ. Shape of the traveling pulse profile also depends on λ but also on a complex combination of parameters in the stationary-moving transition

probabilities $p_{sm}$, $p_{ms}$. For more detail on how the model depends upon parameters, see Parameter sensitivity analysis.

Qualitative and quantitative observations suggest that the tail of the density distribution of a hopper band is roughly exponential in shape [4, 9, 10]. Results from our agent-based model agree. We fit an exponential curve $e^{a+bx}$ to the tail of our average traveling pulse and obtained $a = 4.11$, $b = 0.2831$ and a root-mean squared error of 15.94, see the gold curve in Fig 4B. These data are within an order of magnitude of those observed in the field from Figures 1 and 2 in [9]. (To make this comparison, one must convert the independent variable in the exponential from space in our numerical data to time in the empirical data. Since the pulse travels with with constant speed $c$, we have $x = ct$ and our converted exponential is $e^{a+bct}$ with $bc = 1.50 \times 10^{-3}$, compared with exponential rates on the order of $10^{-2}$ in [9]).

### Theoretical results for the PDE: Hopper bands as traveling waves

**Hopper bands require $R$-dependent switching.** To demonstrate the importance of the $R$-dependence in the switching rates $k_{sm}$, $k_{ms}$, we first consider a simplification of our model. Suppose that these switching rates are constant ($k_{sm} \equiv \alpha$, $k_{ms} \equiv \beta$). We mathematically determine the long-time behavior of solutions to this simplified problem in S1 Appendix. For any locust density solution $\rho = S + M$, the center of mass moves to the right with a speed that approaches $v\frac{\alpha}{\alpha+\beta}$ as $t \to \infty$. This is consistent with our search for traveling-wave solutions. However, we also find that the asymptotic standard deviation $W_\sigma \sim \sqrt{t}$ so that solutions spread diffusively for all time. In other words, no coherent hopper bands form in the long-time limit. Gray dashed lines in Fig 5A depict this behavior, illustrating the decay of a locust density profile with resource-independent switching rates.

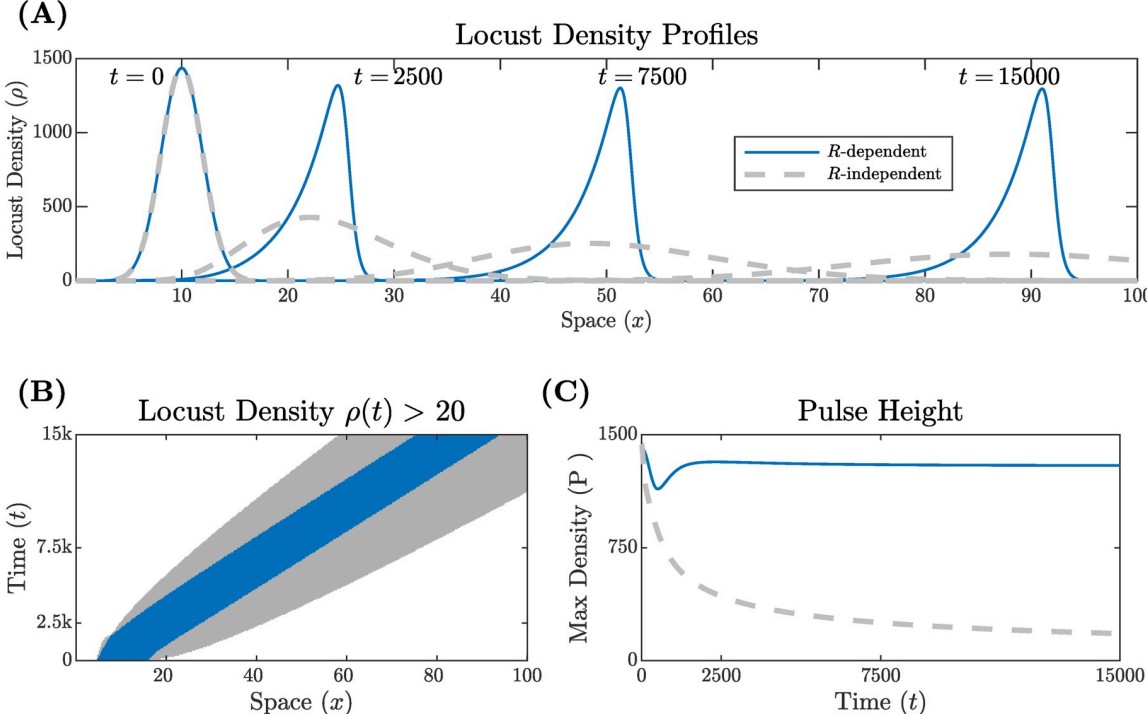

**Fig 5. Locust density profiles with $R$-dependent (solid blue line) and $R$-independent (dashed gray line) switching rates.** Each profile evolves from the same initial condition. (A) Shows snapshots of the density profiles over distance and time for both types of switching rates. (B) Illustrates the width of the bands where color represents a locust density greater than 20. (C) Displays the peak density of each pulse over time.

**Existence of traveling wave solutions.** Returning to the main case with $R$-dependent switching, we show existence and development of hopper bands as traveling wave solutions to the PDE (9). A traveling wave is a solution with a fixed profile that propagates right or left with a constant speed $c$. Since locusts move only to the right in our model, we expect right-moving traveling waves and S2 Appendix includes a mathematical analysis of these solutions. Numerical simulations suggest that these traveling waves organize all long-time dynamics of the model. That is, all solutions with our initial condition appear to converge to a traveling wave. Biologically, we conclude that a typical initial distribution of locusts aggregates into a coherent hopper band. The solid blue curves in Fig 5A show snapshots of the asymmetrical traveling wave created by $R$-dependent switching rates.

Fig 5B and 5C compare the width and maximum density of the profiles for switching rates with and without resource dependence. In Fig 5B, colored regions correspond to a locust density greater than 20 locusts/m with gray and blue corresponding to $R$-independent and $R$-dependent switching rates, respectively. As the locust band without $R$-dependent switching progresses, the width of the gray region increases in time, showing diffusive spreading. On the other hand, the width of the locust band with $R$-dependent switching (blue) remains constant over time. Additionally, the locust band with $R$-dependent switching reaches a constant height as seen in Fig 5C (blue). In contrast, the maximum locust density with $R$-independent switching rates decreases over time as locusts spread out (dashed gray).

**Traveling waves dynamically select collective observables.** By viewing hopper bands as traveling waves, our existence proof also determines a relationship between the total number (or total mass) $N$ of locusts in our 1-meter cross section, the average band speed $c$, and the initial and remaining resources $R^+$ and $R^-$. In S3 Appendix we show that these four variables must satisfy an explicit equation for any traveling wave. One consequence is that our model exhibits a selection mechanism whereby the average band velocity and the remaining resources are determined by the number of locusts in the band and the initial resource level.

These explicit equations are illustrated in Fig 6. Each subfigure shows curves on which $R^+$ is constant (level curves). Plotting these in the $N$, $c$-plane (mass vs. speed), we obtain Fig 6A. (Here each curve is parameterized by $R^-$.) Note that the curves appear monotone: speed $c$ increases as a function of mass $N$. Biologically, this is what one expects; a larger swarm

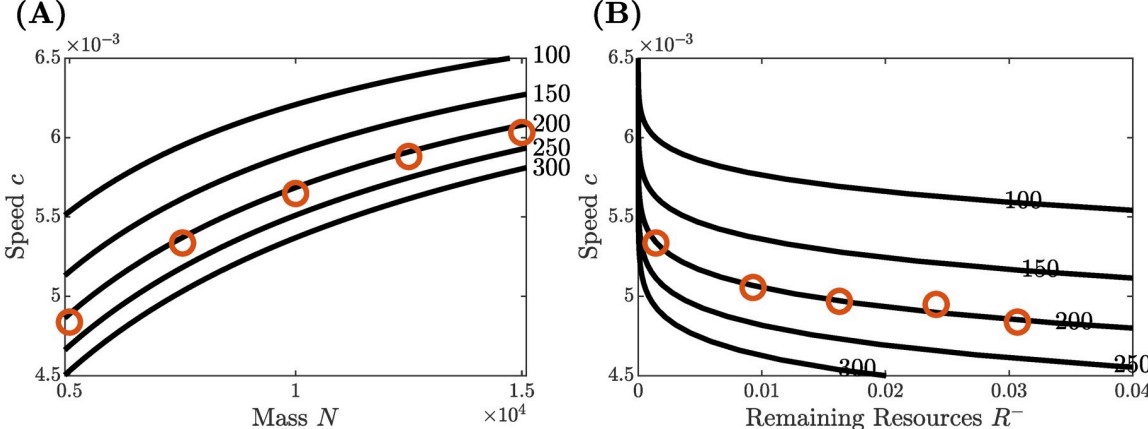

**Fig 6. Level curves on which initial resources $R^+$ are constant (black curves) computed explicitly from the analytic formulas of the PDE model, and numerical data points (orange circle) generated by direct simulation of the ABM.** (A) Illustrates that the average swarm speed $c$ monotonically increases with the mass $N$ and shows agreement with numerical data obtained for $N = 5000, 7500, 10000, 12500, 15000$. (B) Illustrates the inverse relationship between $c$ and $R^-$ and shows agreement with numerical data for $N = 5000, 5250, 5500, 6000, 7500$.

consumes food more quickly and moves on at a faster average pace. In Fig 6B, we plot the same level curves in the $R^-$, $c$-plane (remaining resources vs. speed). (Now each curve is parameterized by $N$.) Again the curves are monotone but we now see that speed $c$ decreases as a function of remaining resources $R^-$. Here we also observe that the speed is much more sensitive when the remaining resources are very small. In S3 Appendix, we use the explicit formulas to prove the monotonicity of speed as a function of input parameters.

## Agreement between ABM and PDE

We evaluate agreement between our two models by comparing the collective observables of Table 3. We divide these into two groups: the shape of the band as characterized by maximum density P, width $W$, and skewness $\Sigma$; and the mean speed $c$ and remaining resources $R^-$, which we consider to be more agriculturally relevant.

**ABM simulations and PDE analysis.** The quantities $c$ and $R^-$ can be determined for the PDE model via the traveling wave analysis of the last section. This analysis results in explicit formulas in S3 Appendix. Substituting input parameters total mass $N$ and initial resources $R^+$, one can calculate exact results for $c$ and $R^-$. These relationships are represented by level curves in Fig 6, for details see Theoretical results for the PDE: Hopper bands as traveling waves.

We ran direct numerical simulations of the ABM for selected values of the total mass $N$ = 5000, 5250, 5500, 6000, 7500, 10000, 12500, 15000. In each simulation we used our example values for all other biological parameters. We ran each simulation for $2.5 \times 10^4$ time steps with $\Delta t = 1$ for a final end time of 25000 sec and confirmed that the simulation reached the end of transients. We measured the collective speed $c$ and remaining resources behind the band $R^-$ for each simulation. The resulting values agree with the explicit formulas to within 1% and are shown in Fig 6 (orange circle).

**Direct simulation of both models.** We used direct numerical simulation of both models to evaluate their agreement on the basis of the shape characteristics maximum density P, standard deviation width $W_\sigma$, and skewness $\Sigma$.

We ran both models for $nt = 2 \times 10^5$ time steps using our example parameters and a range for the foraging rate $\lambda$ so that $-8 < \log(\lambda) < -4$. For each value of $\lambda$, we plot the shape characteristics in Fig 7. For the PDE, we measure the shape characteristics of the final output density profile. For the ABM, we measure the shape characteristics of a time-averaged density profile (as constructed in Fig 4B). The plots in Fig 7 are the result of continuation in the parameter $\log(\lambda)$. We begin with $\log(\lambda) = -4$ and chose initial conditions computed from independent simulations of each model. For each value of $\log(\lambda)$ the algorithm proceeds as follows: We run both models for $nt$ time steps; measure P, $W_\sigma$, and $\Sigma$; choose new spatial grids for each model based on the value of $W_\sigma$; increase $\log(\lambda)$ by 0.1; and use the current output as the next initial condition. Practically speaking, the interval $-8 < \log(\lambda) < -4$ is in fact covered by three such continuations originating at $-4$ and $-7$. Note that our numerical scheme begins to reach its limits as $\log(\lambda)$ approaches $-8$ because there the evolution of the profile shape is so slow that it requires very long computation times to reach equilibrium. This is also why we do not cover the full range of $\log(\lambda)$ explored in the next section.

To visually compare the profiles, see Fig 8. These six profiles are the result of running each model for $nt = 10^6$ time steps with our example parameter values and selected $\log(\lambda) = -7.4$, $-6.3$, $-4.2$. First, note the strong agreement along each row. Second, a data point (gold dot and x) from each of these $\log(\lambda)$ values is included in the plots of Fig 7. Since there is little difference between these data points and the rest in the figure, which are the result of only $2 \times 10^5$ time steps, we can conclude that the shape characteristics have reached near-equilibrium values. The gold x at $\log(\lambda) = -6.3$ demonstrates the stochasticity of the ABM—the maximum of a

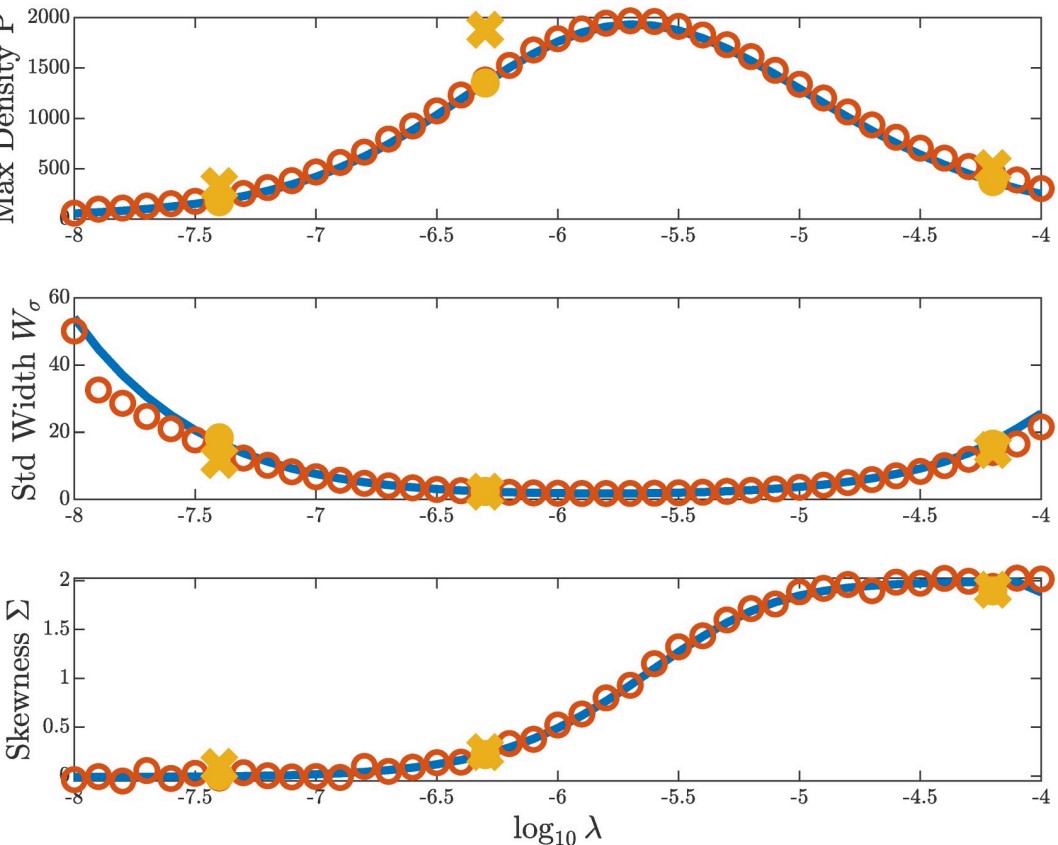

**Fig 7. Comparison of the peak, width, and skewness of profiles from the PDE (blue line) and the ABM (orange circle), both obtained through direct numerical simulation for $2 \times 10^5$ time steps.** Each shape observable is measured from the final numerical output for the PDE and from a time-averaged output the ABM. Longer simulations with $10^6$ time steps, for ABM (gold x) and PDE (gold dot) show little evolution in the profile for longer times. Note that the maximum density is higher for long simulations of the ABM (gold x) because these represent a single instance, rather than an average.

single distribution is larger than the maximum of the time-averaged profile, see Fig 8 (right, center).

Finally, these profiles also provide insight into the possible shapes of density profiles far from our example value of $\log(\lambda) = -5$. Immediately, we notice that the remaining resources $R^-$ behind the pulse decrease quickly as foraging rate $\lambda$ grows, confirming intuition. Next, the shape also varies dramatically as can be seen by noting that the horizontal axes in each row have vastly different scales. In particular, the profiles in the top row are short and wide while the middle row is narrow and tall, all having the same total number of locusts. The bottom row reveals a transition where the resources are nearly all depleted behind the pulse, leading to wide asymmetrical profile as observed in the field [10].

We carry out a more rigorous study of how the model responds to changes in the input parameters in the next section.

## Parameter sensitivity analysis

The sensitivity of the model to its parameters was examined by computing Sobol indices [48] for several biologically observable quantities (see Table 3) with samples from the parameter

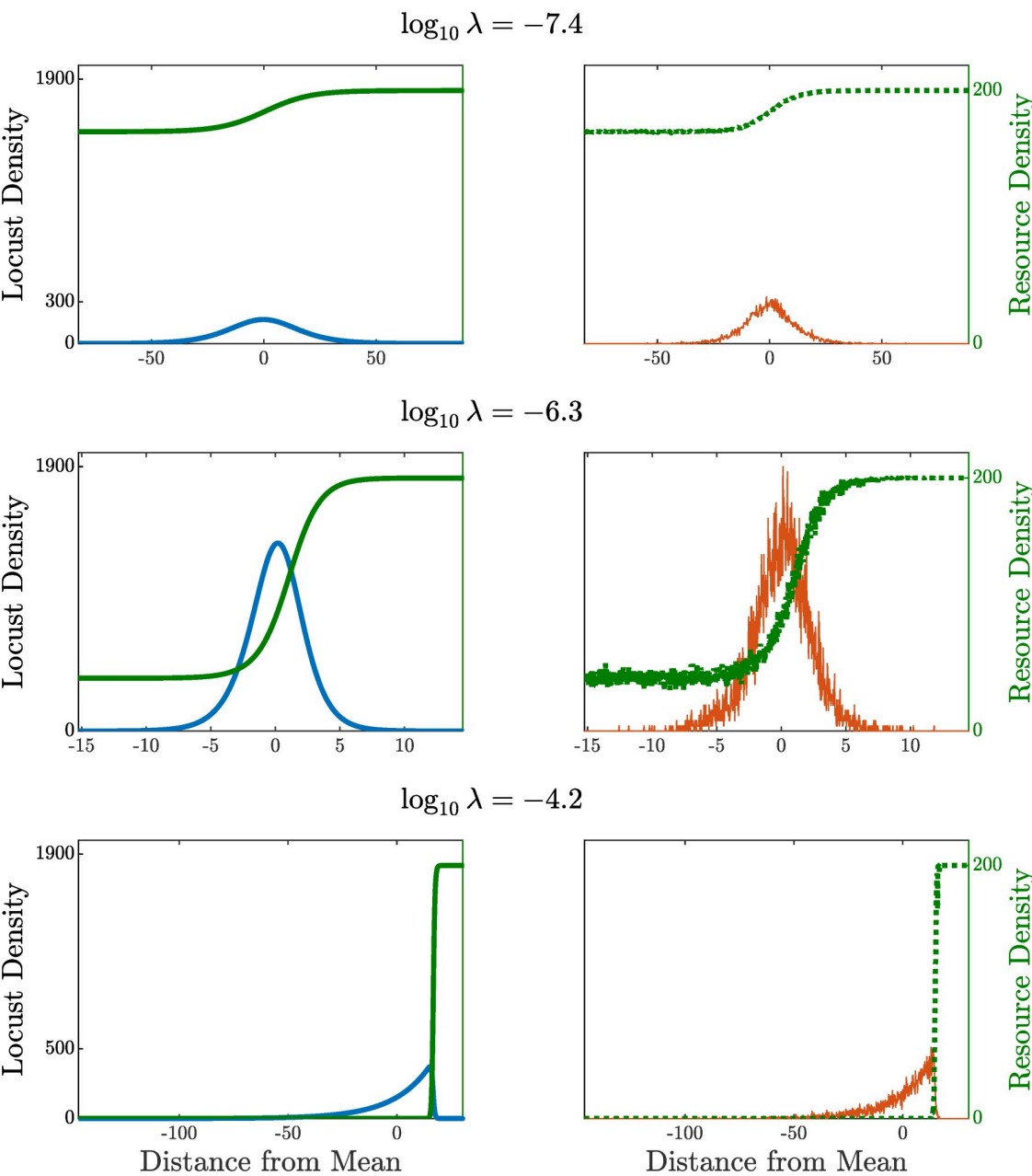

**Fig 8. Model outputs from direct numerical simulation for $10^6$ time steps.** Density profiles from the PDE (blue, left) and histograms from the ABM (orange, right) for selected foraging rates $\log(\lambda) = -7.4, -6.3, -4.2$. For quick visual shape comparison, all outputs shifted so that center of mass is $x = 0$. Each plot corresponds to a data point in Fig 7 (gold x for ABM, gold dot for PDE). Note the differences in scale on the horizontal axes in each plot.

space chosen via Saltelli's extension of the Sobol sequence [49, 50]. Sobol indices represent a global, variance-based sensitivity analysis for nonlinear models that has become extremely popular in recent years for examining the performance of mathematical models in the context of data (e.g., [51, 52]). One of its strengths is the ability to calculate not just first-order (one-at-a-time) parameter sensitivity, but also second-order (two-at-a-time) and total-order (all possible combinations of parameters that include the given parameter) indices [50]. All indices are

normalized by the variance of the output variable. Here, we will focus on the first-order and total-order indices, and note that the presence of higher-order interactions between the parameters can be inferred by comparing differences between these two.

Scalar output quantities for our model (our collective observables) were all chosen with respect to the asymptotic traveling wave solution of the PDE model and are calculated by solving analytically for this solution. The observables chosen are the speed of the traveling wave $c$, the density of remaining resources $R^-$, the peak (maximum) density of the wave profile, the width of the profile measured by its standard deviation $W_\sigma$, and the skewness $\Sigma$ of the profile. Table 3 provides physically relevant ranges for these observables from empirical studies.

In the case of switching parameters $\alpha$, $\beta$, $\theta$, and $\eta$, sensitivity to the ratios $\theta/\beta$ and $\eta/\alpha$ and the ratio difference $\Delta = \alpha/\beta - \eta/\theta$ were used rather than the parameters themselves. One reason for this choice is to guarantee existence of a traveling wave solution; existence is guaranteed whenever $\Delta > 0$. Note that we also would like $\eta/\alpha < 1$ and $\theta/\beta > 1$ so that $k_{sm}$ is a decreasing function of resources $R$ and $k_{ms}$ is an increasing function of resources. Additionally, these two conditions imply that $\Delta > 0$ so there is consistency between these constraints. Another reason for using these ratios lies in mathematical interpretation: $\Delta$ is a measure of the difference in asymptotic switching rates behind the pulse (small $R$, $\alpha/\beta$) and ahead of the pulse (large $R$, $\eta/\theta$), and the two other ratios $\eta/\alpha$ ($\theta/\beta$) describe how much the stationary to moving (moving to stationary) switching rates depend on $R$. More specifically, as these ratios approach 1 the switching rate changes little as $R$ increases, while $\eta/\alpha$ close to zero or $\theta/\beta$ large implies a relatively large change in the switching rate. With these ratios and a value for $\beta$ (chosen because we have some biological data for $\beta$), all four parameters in the ratios are uniquely determined.

Results are shown in Fig 9. All bars are stacked with each color corresponding to a different observable; reading across the parameters, the length of like colors can be compared. Critically, the parameter sensitivity is with respect to the range of parameter values given in the table included with Fig 9. These ranges were chosen to represent both biologically expected values (when information about these values could be obtained) and the necessary conditions for a traveling wave solution.

One immediate observation concerning the Sobol sensitivity analysis in Fig 9 is that $\log_{10}(\lambda)$ and $\log_{10}(\Delta)$ have a large effect on the collective observables of the pulse. Recall that $\lambda$ is the parameter encoding the foraging rate; $\Delta$ is discussed in detail earlier in this section. The bottom row of Fig 9 shows the impact of these parameters on the density of resources asymptotically left behind the locust band as a fraction of the starting density ($R^-/R^+$) and the ratio of the traveling wave velocity to the marching speed of a locust ($c/v$) respectively. Max density, pulse width as measured by standard deviation, and skewness also depend heavily on these two parameters as seen in the top row of Fig 9. This is in fact unsurprising since $\lambda$ and $\Delta$ have by far the largest sample space range in terms of order of magnitude, and for this reason are the only ones examined on a log scale while all other parameters are on a linear scale. To explain this discrepancy, we remind the reader that our chosen sampling ranges represent our uncertainty about the value that the parameters should take on in nature given all the information we were able to find in the biological literature. Our conclusion with this analysis then is that the model is in fact sensitive to this level of uncertainty in $\log_{10}(\lambda)$ and $\log_{10}(\Delta)$, and we should seek to narrow down the possibilities given what we know about observable, biological characteristics of the traveling locust band generated by our parameter choices (Table 3). Through the following numerical analysis of our sample data, we do just that.

To begin, we further illustrate the effect of varying $\log_{10}(\lambda)$ and $\log_{10}(\Delta)$ on the fraction of resources remaining $R^-/R^+$ (in Fig 10A) and on the ratio of the average pulse speed compared to the speed of a moving locust $c/v$ (in Fig 10B). In each figure, we plot a uniform random subset of the sample points used in the Sobol sensitivity analysis for the purpose of down-sampling

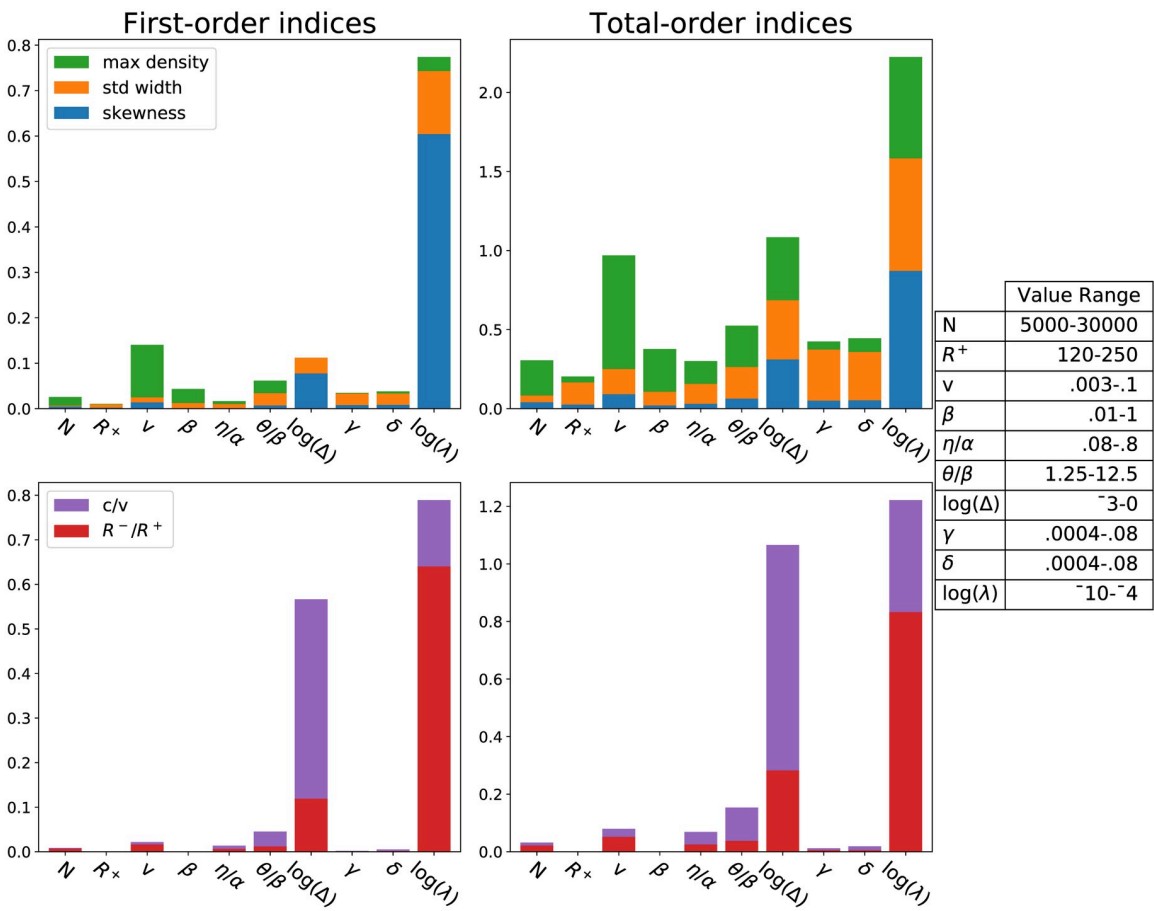

**Fig 9. Sensitivity of various traveling wave observables to model parameters (bars are stacked).** See Table 2 for parameter definition and ranges; this analysis was run using 4,400,000 samples from the given ranges. All log functions are base-10. First-order indices neglect all interactions with other parameters while total-order indices measure sensitivity through all higher-order interactions. Max 95% confidence intervals for the response variables was 0.01 for the first-order indices, 0.049 for total-order.

the image and better visualizing sparse regions in the parameter space—it is qualitatively the same when using all sample points from the Sobol analysis.

Inspecting Fig 10A we note that, generally, at small λ a majority of resources persist after the locust front has passed while at large λ, the majority of resources are consumed. The red dot on the λ axis represents the example parameter set described in Table 2 which we believe to be a relatively feasible choice of parameter values in the context of the biological data about the observables in Table 3. We acknowledge that this appears to suggest the locusts leave behind no vegetation at all, but remember that our variable $R$ represents locust-edible resources—there may be dry plant matter left behind that even locusts would not consume.

Locust swarms observed in the field have a characteristic sharp rise at the beginning of the front and an exponential decay in the tail, see [9] for a quantitative analysis. This observation suggests that the skewness $\Sigma$ is positive and less than or approximately equal to 2 (see Table 3). Fig 11 investigates the relationship between skewness $\Sigma$, foraging rate λ, and the difference of ratios $\Delta$. For $\lambda < 10^{-7}$, most values of $\Sigma$ are negative, indicating an unrealistic density profile leaning to the left. As λ increases from $10^{-7}$ to $10^{-4}$, $\Sigma$ increases and clusters around 2. A smattering of points appear with $\Sigma > 2$ but these all correspond to profiles with unbiologically large

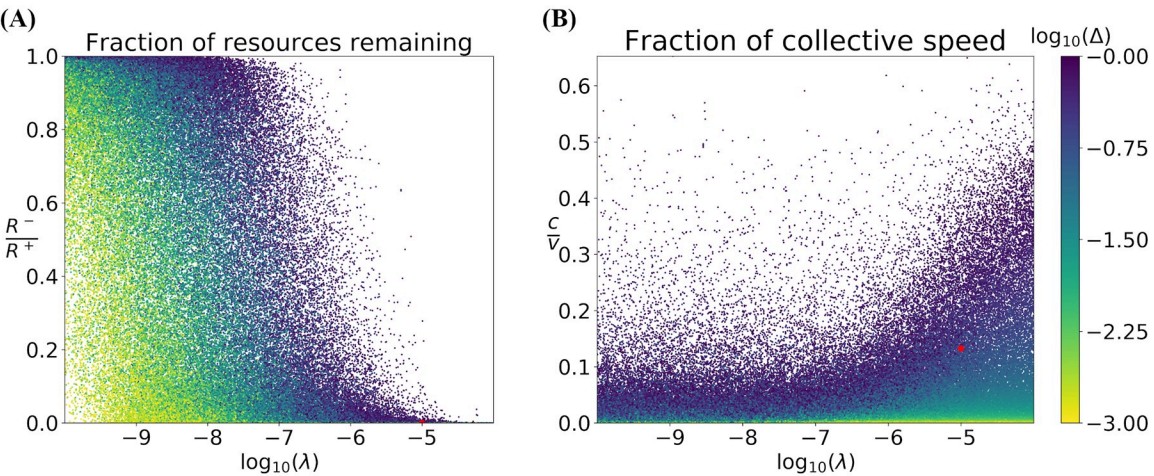

**Fig 10. Scatter plot of (A) remaining resource fraction $R^-/R^+$ and (B) fraction of the traveling wave speed $c$ over individual locust speed $v$ as a function of the foraging rate $\lambda$ and colored by $\Delta$, the difference in asymptotic switching rates behind and ahead of the pulse.** Points are taken from the parameter ranges in Table 2 and represent 5% of all the points sampled for the Sobol analysis, chosen randomly. The red dots represent the example parameter set described in Table 2.

maximum locust densities, as demonstrated by Fig 11B which only shows profiles with maximum locust density <10,000.

To identify a set of parameter inputs that would produce a density profile with observable quantities matching those found in the literature (see Collective observables—model outcomes), we finally sorted the data underlying these figures and conditioned on desirable observable properties as specified in Table 3. This resulted in the example parameters specified in Table 2, with context provided by Figs 10 and 11. The results of the model run with these parameters can be seen in the figures included within the previous results sections.

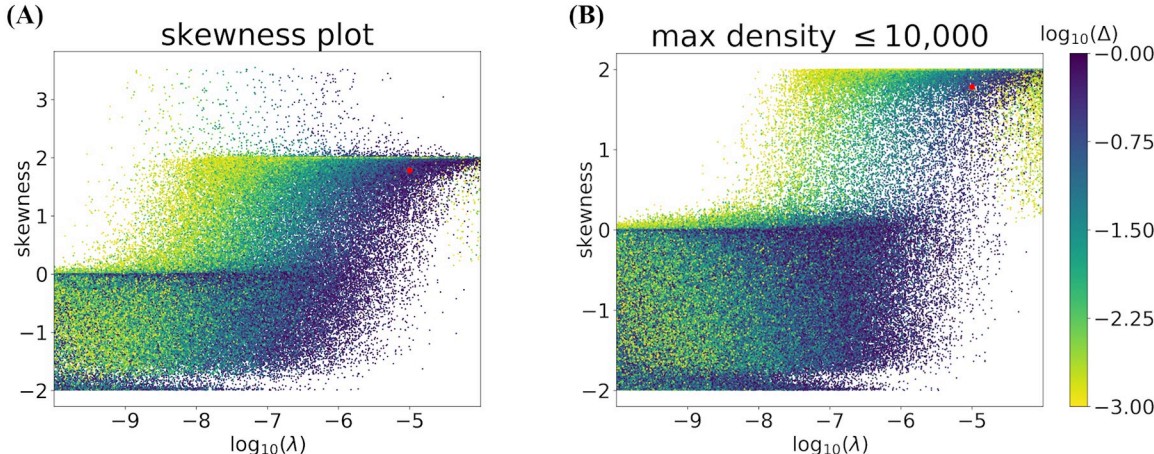

**Fig 11. Skewness as a function of foraging rate ($\lambda$) and colored by $\Delta$.** Fig 11A is representative of the entire sampled parameter space while Fig 11B shows only the points with peak wave amplitudes less than 10,000 locusts per square meter. Points are taken from the parameter ranges in Table 2 and represent 5% (in the case of Fig 11A) and 50% (in the case of Fig 11B) of all the points sampled for the Sobol analysis, chosen randomly. The red dot represents the example parameter set described in Table 2.

## Discussion

We present two minimal models for hopper bands of the Australian plague locust and demonstrate that resource consumption can mediate pulse formation. In these models all locusts are aligned and are either stationary (and feeding) or moving. Our agent-based model (ABM) tracks the locations, state, and resource consumption of individuals. In tandem, our partial differential equation (PDE) model represents the mean-field of the ABM. Both models generically form pulses as long as the transition rate from stationary to moving states is diminished by the presence of resources and/or the transition from moving to stationary states is enhanced by the presence of resources.

The ABM and the PDE each allow us to examine different facets of the problem. The ABM is easy to simulate and directly relates to observations at the scale of individual locusts. It captures pulse formation and propagation, reproduces the stochastic variation seen in the field, and lets us track individual locusts which perform random walks within the band. The PDE model provides a theoretical framework for proving the existence of traveling pulses. This framework facilitates analysis of the collective behavior of the band including mean speed, total resource consumption, maximum locust density, pulse width, and pulse skewness. In turn, this theory enables us to conduct an in-depth sensitivity analysis of the pulse's characteristics with respect to the input parameters. The two models are consistent in the following sense: the characteristics of pulses in the ABM, when averaged over many realizations, correlate precisely with the densities in the PDE model.

We are fortunate that there is a healthy literature addressing the behavior of the Australian plague locust, notably the shape and speed of observed bands [4, 9, 10, 16, 46]. We have used these studies to estimate ranges for the organism-level parameters in our models. Some of these parameters (such as individual marching speed) have been carefully measured yielding narrow ranges. Others (notably the individual foraging rate) can only be deduced to lie within a range of several orders of magnitude. Using these biologically plausible ranges, we analyze the sensitivity of a pulse's characteristics to changes in the input parameters. Sampling parameter values from within these ranges, we examine the resulting speed, remaining resources, and pulse peak, width, and skewness of over $4.4 \times 10^6$ traveling pulse profiles. Sobol sensitivity analysis quantifies the change in these characteristics as a function of the change in each input parameter. Guided by this analysis, we are able to identify a set of parameters that produces pulses concordant with those observed in the field. We conclude that resource-dependent transitions are a consistent explanation for the formation and geometry of traveling pulses in locust hopper bands.

A reasonable question is whether a different mechanism can drive pulse formation or if the formation of pulses is enhanced by a combination of behaviors. Prior works, both for the Australian plague locust and for other locust species, investigate a variety of social mechanisms for collective movement in hopper bands. In two agent-based modeling studies [23, 27], pulses are among a handful of aggregate band structures obtained by varying the parameters that model individual locust behavior. A continuum approach in [30] finds traveling pulses in a PDE similar to our Eq (9) but without accounting for resources. Instead, social behavior is encoded via dependence on locust density of both the transition rates and the speed. This is coarsely akin to our model where resource-dependent transitions between moving and stationary states is necessary for pulse formation, see S1 Appendix. However, a model with social behavior as the only driving factor does not account for the observations of Clark [4] and Hunter [10] that *C. terminifera* manifests pulse-shaped bands with varying shapes based on the surrounding vegetation. We believe that incorporating both social and resource-dependent behaviors will better reproduce field observations.

Further experiments and field observations could help to elucidate the combined roles of resources and social behavior in the formation of hopper bands. While there is a considerable literature on the social [3, 9] and feeding [15, 46] behavior of *C. terminifera*, less progress has been made in quantifying the effect of food on individuals in dense bands exhibiting collective motion. One notable exception is the recent study of Dkhili et al. [19]. Looking ahead, field data could be collected as video while a hopper band moved through lush vegetation, see the methods in [9]. With continuing advances in motion tracking, for instance as employed in [28], one could collect time-series data on each individual moving through the frame. From such data one could draw out the effects of nearby vegetation, satiation or hunger, and local locust density on pause-and-go motion. In turn these processes could be modeled more thoroughly.

We see our present models as a testbed upon which one may develop extensions that capture more of the complexity in locust hopper behavior. The most natural of these extensions is to consider locusts' social behavior, as discussed above. A second is to include stochastic, individual, and environmental variation. This could be incorporated into the agent-based model in order to examine the robustness of pulse characteristics with respect to a distribution of individual marching speeds, or even large hops, as in [27]. For the PDE model, random variations in locust movement could naturally be represented by a linear diffusion term. Thirdly, we could incorporate motion of locusts transverse to the primary direction of propagation. This two-dimensional model might aim to capture the curving of the front of hopper bands often seen in the field. Lastly, large changes in resource density could be included to represent the band entering or exiting a lush field or pasture, with a view towards informing barrier control strategies as discussed in [11, 19]. These extensions could help explain the variety of morphologies and density profiles—including curving dense fronts, complex fingering, and lower-density columns—observed in hopper bands of the Australian plague locust and other species.

## Supporting information

**S1 Appendix. Resource-independence: The Telegrapher's Equation.** Supposing that the stationary-moving transition rates $k_{sm}$, $k_{ms}$ are independent of $R$, we construct an argument using moments of the resulting density distributions to show that solutions spread indefinitely with a gaussian shape. In particular, there are no coherent pulse solutions with a steep front.
(PDF)

**S2 Appendix. Traveling wave analysis.** We prove the existence of traveling wave solutions to the PDE (9) using an invariant region argument. The existence result also provides a selection mechanism; that is, for a given set of parameter inputs there is only one traveling wave.
(PDF)

**S3 Appendix. Formulas for *N*, *c*, $R^+$, $R^-$.** In S2 Appendix we show existence of a traveling wave solution. We now characterize this solution with explicit formulas that relate *N*, *c*, $R^+$, and $R^-$. Given any two of these variables and the remaining model parameter inputs, these formulas determine the other two exactly.
(PDF)

**S1 Video. Visualizations of the Agent-Based Model.** Video showing timesteps from a simulation of our Agent-Based Model (ABM): Pause-and-go motion on a space-time grid with example parameter values from Table 2. The top panel shows a schematic of the 1-meter cross section represented by our one-dimensional model. Each locust (maroon and blue dots) has a unique horizontal lane in this schematic; there is no vertical motion. The bottom panel shows

a line plot (orange) reporting the number of locusts, both stationary and moving, at each spatial gridpoint and the resource density (green); compare to Fig 4B.
(M4V)

## Acknowledgments

This collaboration began as part of a Mathematics Research Community of the American Mathematical Society. We thank Tom Barr for his role in organizing this program and also to the leaders of our session on Agent-based Modeling in Biological and Social Systems including Maria R. D'Orsogna, Alan Lindsay, Chad Topaz, Alexandria Volkening, and Lori Ziegelmeier. We are also grateful to the Institute for Advanced Study which hosted our group for a subsequent research visit through their Summer Collaborators Program. We wish to thank Leah Edelstein-Keshet who brought this problem to our attention and for her thoughtful insights on this work. Andrew Bernoff also thanks Edward Green and Jerome Buhl and the hospitality of the University of Adelaide for supporting a visit which lead to many insightful conversations about this work. We are also deeply grateful to Jerome Buhl for his encyclopedic knowledge of locusts, notably the Australian plague locust, for sharing and confirming some of the biological estimates in this paper, and for his comments on an earlier draft of this work.

## Author Contributions

**Methodology:** Andrew J. Bernoff, Michael Culshaw-Maurer, Rebecca A. Everett, Maryann E. Hohn, W. Christopher Strickland, Jasper Weinburd.

**Writing – original draft:** Andrew J. Bernoff, Michael Culshaw-Maurer, Rebecca A. Everett, Maryann E. Hohn, W. Christopher Strickland, Jasper Weinburd.

**Writing – review & editing:** Andrew J. Bernoff, Michael Culshaw-Maurer, Rebecca A. Everett, Maryann E. Hohn, W. Christopher Strickland, Jasper Weinburd.

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
