## [Decision Letter · Decision Letter 0]

8 Dec 2019

Dear Dr Weinburd,

Thank you very much for submitting your manuscript 'Agent-based and continuous models of hopper bands for the Australian plague locust: How resource consumption mediates pulse formation and geometry' for review by PLOS Computational Biology. Your manuscript has been fully evaluated by the PLOS Computational Biology editorial team and in this case also by two independent peer reviewers. The reviewers appreciated the attention to an interesting question and both praised the theoretical modeling, but raised some concerns about the manuscript as it currently stands. In particular, Reviewer 2 raised substantial concerns abut the biological underpinnings of the modeling assumptions and, implicitly, about the biological significance of the predictions, that need to be thoroughly addressed. While your manuscript cannot be accepted in its present form, we are willing to consider a revised version in which the issues raised by the reviewers have been adequately addressed. We cannot, of course, promise publication at that time.

Sincerely,

Corina E Tarnita

Associate Editor

PLOS Computational Biology

Natalia Komarova

Deputy Editor

PLOS Computational Biology

[LINK]

Reviewer's Responses to Questions

**Comments to the Authors:**

Reviewer #1: I greatly enjoyed reading this paper and. Commend the authors for a truly fine job. First, a disclosure; I am a coauthor on other locust papers with Prof Bernoff, so I am not fully at arm's length. My name also appears in the Acknowledgements, as the editor could easily see. However, I do not feel that this has affected my objective assessment of this paper. In fact, I am all too aware of the challenges that the authors had to overcome, most particularly with estimating parameters and finding the reasonable range of values, as well as assembling a mode that has important factors, without overwhelming details.

I think this paper is an excellent example of rational and well-thought out study that combines biologically determined facts and data from the biological literature with a satisfying and convincing set of models. The authors did it all: setting up an agent-based model, deriving the corresponding mean-field PDE, estimating the microscopic parameters, comparing the observed group behaviour to biologically observed ones, and conducting a detailed (Sobol) parameter sensitivity analysis.

They show how the locust marching speed and density profile are governed by the foraging rate, the resource availability, and the transition rates between moving and stationary feeding locusts.

A great feature of this paper is also that it can suggest specific experiments that could be used to refine and/or improve the model accuracy. Indeed, one of my suggestions is to make this more explicit in the Discussion, where a paragraph could be added to list the top few field measurements or lab data that could help in this regard.

I believe that this paper is idea for PLOS CB, and that it will be useful to a broad readership, and not just those who are interested in locusts. I recommend publication after a few minor issues are addressed.

Signed: Leah E-K

Minor points

There have been previous models for bands of animals moving into a resource gradient(s). Perhaps these and a few others should be cited, and some general comments made to distinguish the math from what is done here. Examples include (but are not restricted to):

Gueron, S. and Liron, N. 1989, A model of herd grazing as a travelling wave, chemotaxis and stability, J. Math. Biol. 27: 595}608

Novick-Cohen, A. and Segel, L. A. 1984, A gradually slowing travelling band of chemotactic bacteria, J. Math. Biol. 19: 125}132

In a lot of these, the animals are sensing a gradient of the resource to navigate by, while in this paper, the locusts all "know" to just march up the x axis. This could be discussed somewhere as it has some important implications, in particular for individuals far form the front.

The paragraph between L22-38 could be organized a bit better, as it starts with hoppers and then gets into the egg hatching later on.

L 109: In an agent-based model, you can use a distribution of hopping speeds. Can the authors speculate how this would affect the outcome? Is data for the speed distribution available?

In fact, it would be good to mention (in Sec 1.1) how many of the assumptions are based on empirically measured parameters or distributions.

There are some aerial photos of the Australian hoppers. It would have been nice to include one, perhaps obtained from Buhl or other Australian agency? Also, it would be nice to show one simulation in a full 2D geometry with locusts moving along radii of their expanding bare-patch, or along the route of shortest distance to the front. On L 139 the authors claim this social interaction is essential in 2D, but I am not certain that this is absolutely the case. (Could a 2D simulation help argue this point?)

The transition rates in Eqn (2) make sense. But why not start with (3) and work forwards? Why get to (3) by "working backward"?

There is a bit of repetition between sec 1.2 and 1.3 that could be reduced somewhat. In fact, some of the details of sec 1.3 could almost be put in the SI.

Identifying parameters for a model is always a challenge, and the authors do this very well. In some cases, they use data for the desert locust S. gregaria to estimate transition rate parameters. How similar of different are these two species? Are we comfortable with borrowing those parameters from African locusts for the Aussie locusts? What experiments should be done to get a better set of estimates for the Australian locust?

L347 Please clarify feeding vs foraging.

L430 - 434 I recommend adding the values listed here to the parameter table 3, rather than having them in this paragraph.

It is possible that Fig 4 could be combined into a single panel.

Have the authors considered comparing Fig 4 to data published previously? It would seem to be a natural thing to do in such a figure.

L494 A traveling wave could propagate either right or left.

The authors assume that locusts are either stationary or moving at a constant speed in 1 D towards the resource. In this way they can get a traveling wave without losing locusts that stray away in the back. This may be a result of assuming no random motion at all, and no distribution of locust speeds. Could the authors please comment on (a) the realism of this simplification and (b) whether adding some diffusive motion would affect the existence and/or speed of the TW. Also, how do locusts far from the front know which way to go?

Fig 7 caption: an time-averaged output the ABM.

 a time-averaged output of the ABM.

Fig 9: is it needed to repeat the table on the RHS? It appears to be identical in both rows.

Would it be reasonable to combine Figs 10 and 11 into a single figure with 2 panels? (Similar to fig 12).

Reviewer #2: The manuscript presents two models, one agent-based and the second a PDE model, which describe the formation of advancing bands of locusts. The two models are one-dimensional and are essentially the same (the PDE is a local average of the agent-based one). The paper is clear and well written. Both numerical simulations and analytical derivations (of standing wave solutions for the PDE) are described.

Unfortunately, I cannot recommend publication of the paper at PLoS Comput Biol, at least not in the current form of the manuscript, due to several fundamental difficulties with the assumptions underlying the models, as described below.

1. Does local food abundance indeed regulate the propensity of the locust to move or stand? This idea has been recently tested experimentally by Dkhili et al, “Effects of starvation and vegetation distribution on locust collective motion”. Journal of Insect Behavior, 1-11 (2019). In particular, the authors find that, while starved animals stop and eat before initiating marching activity, ”The results did not show any effects of vegetation distribution on the oriented motion and mean speed”. In other words, feeding and marching are not coupled – starved animals ate first and only then marched, at least in the desert locust studied by Dkhili et al. In contrast, line 33 of the current manuscript states that “While marching, they consume large quantities of green biomass” and cite [11]. Looking through [11], I did not find any report indicating that the Australian plague locusts feed during marching. However, I would be happy to be corrected. Therefore, the authors should, in the very least, make it clear that the main assumption underlying the model, that the transition probabilities p_{sm} and p_{ms} depend on R, are hypotheses that have not been verified experimentally for the Australian plague locusts or are (at best) species specific (and provide a reference).

2. Do standing animals always eat? To the best of my knowledge this is not true, neither in the lab, nor in field studies. See again, Dkhili et al who report a post-prandial resting period for dessert locusts. Thus, the stop-and-go movement pattern may only be interpreted as coarse-grained behavior of moving and standing+eating periods. In particular, the parameter fit from arena experiments without food may not be appropriate here.

3. Why is the food consumption rate proportional to R? The authors hypothesize that when food concentration is low, the locust need to spend time searching for it (in the direction perpendicular to motion, which is not modelled explicitly). However, the model assumes a preferred direction, i.e., the locust “know” that the food is at the front. Therefore, why search sideways? Similarly, how do locust “measure” R (to regulate the standing<->moving transition rates)?

4. Lack of interaction between the species. It is well documented that the propensity of locusts to move and align their direction with others is highly dependent of their conspecifics. While the authors allude to that in the discussion, it is not clear why this basic property, which has been suggested to be pivotal to locust collective motion [35, 1 and many more] is ignored.

Minor comments:

1. Problem with the sentence on line 41.

2. Line 249: The scheme has to compute derivatives upstream.

3. Fig 1: Is this taken from experiment?

4. Fig 3: It is instructive to note that the mean speed is simply vM/N.

5. I suggest discussing the effect of noise/diffusion in the discussion section.

To summarize, I find that in its current form, the models studied in the manuscript are interesting, but only from an academic perspective on the theory of collective motion in general. Unless the authors can provide adequate explanations to the comments above, the models may not applicable to realistic locust swarms.

**Have all data underlying the figures and results presented in the manuscript been provided?**

Reviewer #1: Yes

Reviewer #2: Yes

PLOS authors have the option to publish the peer review history of their article (what does this mean?). If published, this will include your full peer review and any attached files.

Reviewer #1: Yes: Leah Edelstein-Keshet

Reviewer #2: No

---

## [Decision Letter · Decision Letter 1]

23 Mar 2020

Dear Dr. Weinburd,

We are pleased to inform you that your manuscript 'Agent-based and continuous models of hopper bands for the Australian plague locust: How resource consumption mediates pulse formation and geometry' has been provisionally accepted for publication in PLOS Computational Biology.

Best regards,

Corina E. Tarnita

Associate Editor

PLOS Computational Biology

Natalia Komarova

Deputy Editor

PLOS Computational Biology

Reviewer's Responses to Questions

**Comments to the Authors:**

Reviewer #1: The authors have addressed all my questions. It is a fine paper and should be accepted for publication.

Reviewer #2: The authors have addressed all my comments. It is an interesting, well written paper of clear interest to the readers of PLoS Comput Biol. I recommend it is accepted.

**Have all data underlying the figures and results presented in the manuscript been provided?**

Reviewer #1: Yes

Reviewer #2: Yes

PLOS authors have the option to publish the peer review history of their article (what does this mean?). If published, this will include your full peer review and any attached files.

Reviewer #1: Yes: Leah Edelstein-Keshet

Reviewer #2: Yes: Gil Ariel

---

## [Editor Report · Acceptance letter]

23 Apr 2020

PCOMPBIOL-D-19-01903R1 

Agent-based and continuous models of hopper bands for the Australian plague locust: How resource consumption mediates pulse formation and geometry

Dear Dr Weinburd,

I am pleased to inform you that your manuscript has been formally accepted for publication in PLOS Computational Biology. Your manuscript is now with our production department and you will be notified of the publication date in due course.

With kind regards,

Laura Mallard
